# From Gondwana to the Yellow Sea, evolutionary diversifications of true toads *Bufo* sp. in the Eastern Palearctic and a revisit of species boundaries for Asian lineages

Siti N Othman[1,2], Spartak N Litvinchuk[3], Irina Maslova[4], Hollis Dahn[5], Kevin R Messenger[6], Desiree Andersen[2], Michael J Jowers[7], Yosuke Kojima[8], Dmitry V Skorinov[3], Kiyomi Yasumiba[9], Ming-Feng Chuang[10], Yi-Huey Chen[11], Yoonhyuk Bae[2], Jennifer Hoti[2,12], Yikweon Jang[2]*, Amael Borzee[1]*

[1]Laboratory of Animal Behaviour and Conservation, College of Biology and the Environment, Nanjing Forestry University, Nanjing, China; [2]Department of Life Sciences and Division of EcoScience, Ewha Womans University, Seoul, Republic of Korea; [3]Institute of Cytology, Russian Academy of Sciences, St. Petersburg, Russian Federation; [4]Federal Scientific Center of the East Asia Terrestrial Biodiversity Far Eastern Branch of Russian Academy of Sciences, Vladivostok, Russian Federation; [5]Department of Ecology and Evolutionary Biology, University of Toronto, Toronto, Canada; [6]Herpetology and Applied Conservation Laboratory, College of Biology and the Environment, Nanjing Forestry University, Nanjing, China; [7]CIBIO/InBIO (Centro de Investigação em Biodiversidade e Recursos Genéticos), Universidade do Porto, Vairão, Portugal; [8]Graduate School of Human and Environmental Studies, Kyoto University, Kyoto, Japan; [9]Tokyo University of Agriculture and Technology, Tokyo, Japan; [10]Department of Life Sciences and Research Center for Global Change Biology, National Chung Hsing University, Taichung, Taiwan; [11]Department of Life Science, Chinese Culture University, Taipei, Taiwan; [12]Department of Life Sciences and Systems Biology, University of Turin, Turin, Italy

*For correspondence:
jangy@ewha.ac.kr (YJ);
amaelborzee@gmail.com (AB)

**Abstract** Taxa with vast distribution ranges often display unresolved phylogeographic structures and unclear taxonomic boundaries resulting in hidden diversity. This hypothesis-driven study reveals the evolutionary history of Bufonidae, covering the phylogeographic patterns found in Holarctic bufonids from the West Gondwana to the phylogenetic taxonomy of Asiatic true toads in the Eastern Palearctic. We used an integrative approach relying on fossilized birth-death calibrations, population dynamics, gene-flow, species distribution, and species delimitation modeling to resolve the biogeography of the clade and highlight cryptic lineages. We verified the near-simultaneous Miocene radiations within Western and Eastern Palearctic *Bufo*, c. 14.49–10.00 Mya, temporally matching with the maximum dust outflows in Central Asian deserts. Contrary to earlier studies, we demonstrated that the combined impacts of long dispersal and ice-age refugia equally contributed to the current genetic structure of *Bufo* in East Asia. Our findings reveal a climate-driven adaptation in septentrional Eastern Asian *Bufo*, explaining its range shifts toward northern latitudes. We resolve species boundaries within the Eastern Palearctic *Bufo*, and redefine the taxonomic and conservation units of the northeastern species: *B. sachalinensis* and its subspecies.

## Editor's evaluation

Othman et al., resolve the phylogeography of Bufo at global, continental and regional scales. The strengths of the paper reside in the extreme detail, the inclusive nature of the ideas, and insightful reconstructions of the evolutionary history of Bufo species. Furthermore, taxonomic treatments are a strong part and a nice addition of the paper. The manuscript can be of interest to herpetologists, but also biogeographers and evolutionary biologists interested in the geologic history of Asia.

## Introduction

Evolutionary history is critical in explaining the diversification within and between species, and the dynamics between species and environments in ecological zones (*Ricklefs, 2006*). Factors such as past geological events, natural dispersion, and anthropogenic changes (*Dufresnes et al., 2020*; *Othman et al., 2020*) can drive and induce different evolutionary scenarios such as physiological adaptation, genetic variability, and phenotypic divergence (*Luquet et al., 2015*). However, understanding phylogeography and the processes contributing to genetic divergence can be more challenging in taxonomic groups distributed across vast ranges. This difficulty results from the ecological responses to variability in environmental conditions and geographical features (*Zhao and Yu, 2012*). Thus, integrating macroevolution of species groups and microevolution within populations is necessary to understand the evolutionary mechanisms of complex study systems (*Li et al., 2018*).

Amphibians are an excellent model for studying the factors affecting distribution and evolution. Specifically, true toads in the family Bufonidae (bufonids) are well suited to this area of study due to their high species diversity (*Rojas et al., 2018*) and adaptive response to past climate change that led to an evolutionary-recent global radiation (*Van Bocxlaer et al., 2010*). Numerous phylogeographic and systematic studies have characterized the evolutionary diversifications of bufonids, notably in the Holarctic (*Dufresnes et al., 2019*; *Garcia-Porta et al., 2012*; *Pauly et al., 2004*; *Recuero et al., 2012*; *Stöck et al., 2006*), the Neotropics (*Pramuk, 2006*), and in the Western Ghats (*Van Bocxlaer et al., 2009*). However, Asian *Bufo* is generally used as an outgroup (*Garcia-Porta et al., 2012*; *Recuero et al., 2012*) resulting in only a few studies focusing on the spatiotemporal origin of *Bufo* in the Eastern Palearctic. Consequently, the resolution of evolutionary diversification patterns in Asian *Bufo* lineages is limited, and the existing phylogenetic studies are generally geared toward regional sampling and mitochondrial markers (*Borzée et al., 2017*; *Chen et al., 2013*; *Liu et al., 2000*; *Macey et al., 1998*; *Yu et al., 2014*). Although some recent taxonomic revisions have used multi-locus data (*Fong et al., 2020*; *Zhan and Fu, 2011*), the lack of primary fossils calibrations has resulted in molecular dating estimates that mostly depend on paleogeological events and secondary calibrations. Perhaps as a result, contradicting biogeographical hypotheses have been posited regarding the Asian lineage of *Bufo* (*Borzée et al., 2017*; *Fu et al., 2005*; *Igawa et al., 2006*; *Macey et al., 1998*) and inconsistent phylogenetic species boundaries have been proposed (*Macey et al., 1998*; *Liu et al., 2000*; *Fu et al., 2005*; *Pyron and Wiens, 2011*; *Zhan and Fu, 2011*).

To date, three biogeographical hypotheses relevant to the initial emergence of the *Bufo* genus in the Eastern Palearctic have been proposed: (hypothesis 1) the split between a Western and Eastern lineage due to the desertification of Central Asia during the Middle Miocene, c. 12.00 Mya (*Garcia-Porta et al., 2012*); (hypothesis 2) vicariant speciation at the time of the earliest emergence of a high-altitude *Bufo* group distributed in Eastern Tibet, followed by a subsequent dispersal to low elevations in the Late Miocene, c. 10.00–5.00 Mya (*Macey et al., 1998*); and (hypothesis 3) the isolation of an insular endemic clade on the Japanese Archipelago following the drift of the archipelago away from the Eurasian continent in the Late Miocene, c. 6.00 Mya (*Igawa et al., 2006*). Although differing in the details, the hypotheses proposed here agree on the Miocene paleogeological events as key factors in the segregation of the Eastern lineage of *Bufo*.

Comparable to the Western *Bufo*, two or more species of Eastern Palearctic *Bufo* form species complexes with similar morphologies and unclear taxonomic boundaries, such as the *B. gargarizans* and *B. japonicus* complexes (*Matsui, 1986*; *Zhan and Fu, 2011*; *Arntzen et al., 2013*). The divergence of *B. gargarizans* and *B. japonicus* complexes from the other *Bufo* in East Asia, and the radiations within each species complex coincide with the Plio-Pleistocene climatic oscillations (*Fu et al., 2005*), sea-level fluctuations, and ice age glaciations (*Borzée et al., 2017*), and selection pressure in the Anthropocene (*Hase et al., 2013*). Most *Bufo* species distributed on the East Asian mainland are

**eLife digest** The east Asian Asiatic toad (also known by the latin name *Bufo gargarizans*) lives in a wide range of habitats across East Asia including forests, meadows and cultivated land. However, it remains unclear how these toads evolved and became so widespread – partly because it has proved difficult for researchers to clearly define the species and what distinguishes it from other closely-related species of toads (collectively known as *Bufo* toads).

Othman et al. combined several bioinformatics techniques to study Asiatic toads and 38 other species of bufonid toads from across the globe. This approach found that *Bufo* toads first emerged in eastern Asia between 14 and 10 million years ago. This coincides with a point in time when large swathes of land in central Asia turned from adequate to sustain toad populations into desert, suggesting the change in climate prompted the toads to migrate eastwards from central Asia. The *Bufo* toads then divided into two groups of species: one in mainland East Asia and the other in Japan.

Furthermore, the study revealed there is more genetic diversity – that is, more variety in the DNA of individuals – in Asiatic toads than previously thought. The findings also help to define several other species of *Bufo* toads more clearly and describe a new toad species restricted to the Korean Peninsula, northeastern China and the Russian Primorye region: the Sakhalin toad (*Bufo sachalinensis*).

This work demonstrates that a large-scale study of many species across the globe can be used to understand how the species evolved and more clearly distinguish one species from another. The findings of Othman et al. will be of interest to both professional and citizen scientists interested in the natural history of Asia. Furthermore, as several species of *Bufo* toads are in decline in the wild, they provide evidence that may aid future efforts to conserve them.

restricted to the Qinghai-Tibetan Plateau (QTP) and adjacent high-altitude areas. The major exception to this is the *B. gargarizans* complex that is widely distributed in Northeast Asia. The species colonized its current range from a single source through long dispersal on the Asian mainland (*Fu et al., 2005*; *Zhan and Fu, 2011*). Concomitantly, the species expanded and formed a continental lineage of *Bufo* in septentrional East Asia, here defined as the northern-most range of current '*B. gargarizans,*' including clades distributed in the Korean Peninsula and the Russian Far East (*Matsui, 1986*; *Maslova, 2016*). In contrast to the Asian mainland clades, *Bufo* septentrional East Asian clades show a marked impact of Pleistocene glaciations on range shifts (*Borzée et al., 2017*), and possible ice age refugia on the Korean Peninsula, and in the southernmost Amur River Basin (*Borzée et al., 2017*; *Fong et al., 2020*). One of the clades is also likely to have been isolated on the Korean Peninsula since the Last Glacial Maximum (LGM), with limited genetic exchange with other clades. This is plausible because of the Yellow Sea level rise during the LGM (*Li et al., 2016*), as reflected by the monophyly of the Korean *B. gargarizans* clade (*Borzée et al., 2017*).

The taxonomic assignment of East Asian toads using their morphometric and geographic distribution included up to 11 distinctive species in the past (see the chronology of taxonomic updates in *Figure 1—figure supplement 1*). Seven recognized taxa within *Bufo* were assigned to East Asia, ranging from western People's Republic of China (hereafter China) to continental East Asia and associated islands on the continental shelf, and the Amur River Basin. Additionally, four taxa were specific to the Japanese archipelago (*Matsui, 1986*). A systematics revision based on mitochondrial markers then simplified the East Asian mainland *Bufo* to five members, with synonimization of some monophyletic groups and invalidation of paraphyletic taxa (*Liu et al., 2000*). In a more recent multi-locus study (*Pyron and Wiens, 2011*), 10 species were considered valid, although some under sampled taxa remain unresolved (see the type locality distributions and chronology of systematic revision in *Supplementary file 1A*, *Figure 1—figure supplement 1*).

Clade sorting based on morphological and life-history characteristics also influences the tree topology of Asian *Bufo*. Stream-breeders such as *B. andrewsi, B. torrenticola, B. stejnegeri,* and the former '*Torrentophryne*' all demonstrate adaptation to the lotic ecosystem and high-altitude environment (*Liao et al., 2016*). For instance, adults lack tympanums (*Tsuji and Kawamichi, 1996*). Ancestral state reconstruction indicates that the semi-aquatic natural histories of East Asian *Bufo* likely arose independently (*Fong et al., 2020*). Arguably, life-history characteristics may not reflect a phylogenetic relationship, however, such adaptive traits may still help offer explanations in the role of morphological

and ecological characters on the systematics of Asian *Bufo*, and help revisit previous studies in cases of convergence.

Other factors to consider are anthropogenic activities as they have also impacted the current distribution and phylogeny of bufonids, especially in insular regions (*Othman et al., 2020*). An example is an intergradation between the two *B. j. japonicus* subspecies in Japan that were originally geographically isolated from each other. Historically, *B. j. formosus* occurred in the east and *B. j. japonicus* occurred in the west (*Miura, 1995*). This was followed by the invasion of northern Japan (i.e., Hokkaido and Sado) by the former after anthropogenic introductions (*Hase et al., 2013*; *Suzuki et al., 2020*). The distribution of *B. gargarizans* has similarly been strongly influenced by human activities, particularly through the traditional medicine trade (*Zhan et al., 2020*). As a result, contemporary genetic patterns in the species complex show muddled genetic signals (*Lee et al., 2021*). From this, it can be inferred that the past and ongoing trade may lead to the reduction in local diversity within the *B. gargarizans* complex.

The issues related to the taxonomic groupings in the *B. gargarizans* complex are majorly rooted in the inconsistency of the hierarchical systematic ranks used. For instance, the validity of the endemic Taiwan toad, also called Central Formosa toad, *B. bankorensis* as a species was inconsistent across taxonomic revisions despite its position within the *B. gargarizans* clade (*Matsui, 1986*; *Liu et al., 2000*; *Yu et al., 2014*). These inconsistencies confuse not only the interpretations of evolutionary history, but also the ecology. For instance, ecologists have regarded *B. andrewsi* as a species of its own with life-history traits related to adaptation to high altitude (*Liao et al., 2016*), but systematists considered *B. andrewsi* as a junior synonym of *B. gargarizans* (*Fu et al., 2005*). Clearly, an integrative re-evaluation of East Asian *Bufo* species is required. Species delimitation via coalescent methods is a promising tool for addressing these inconsistencies, and a notable improvement to conventional molecular phylogenetic analysis. This methodology has been used to untangle taxonomic issues linked to phenotypic variations and cryptic lineages in other Asian toads such as *Ingerophrynus celebensis* (*Evans et al., 2008*). Additionally, the inclusion in our species tree topology of some previously undersampled bufonids from the eastern tip of Tibet to the northern South East Asia such as *B. pageoti, B. tuberculatus*, a high elevation-restricted *B. gargarizans* subspecies, and *B. gargarizans minshanicus* (hereafter *B. minshanicus*), may increase the robustness of our analyses and resolve the inconsistent placement of these clades within the nomenclature.

The spatiotemporal origin of the East Asian *Bufo* lineage is uncertain and characterized by a limited understanding of the evolutionary processes involved. In addition, species delimitations present a serious taxonomic discrepancy within the *B. gargarizans* complex, resulting in repeated calls for taxonomic clarification at a fine scale. This manuscript is structured into two sections, with different taxonomic and geographic scales (*Figure 1*). The first section focuses on resolving the biogeography of Holarctic bufonid species, following the Bufonidae lineage since the breakdown of the West Gondwana. We use a combination of fossilized birth-death calibrations and multi-locus coalescent-based species tree methods to estimate the most probable time and routes of colonization of bufonids into the Palearctic (*Figure 1*). In the second section, we elucidate the biogeographic pattern of the East Palearctic *Bufo* and resolve the taxonomy of the *B. gargarizans* complex. Here, we evaluate the best species tree topology by testing five alternative hypotheses to recover the taxonomic relationship between Asian species of *Bufo*. We rely on an intensive and widespread sampling, integrated biogeographical analyses, niche differentiation between divergent clades, and model-based species delimitation approach to resolve the taxonomic boundaries of the *B. gargarizans* species complex, including the recently expanded septentrional East Asian clade (*Figure 1*).

## Results

Our study addresses biogeographic scenarios explained in two different sections: (1) the biogeography of Holarctic bufonids and (2) the biogeography of Eastern Palearctic *Bufo* and the taxonomic revision of the species complexes.

### Biogeography of Holarctic bufonids

The goal was to refine the time estimates of the split between clades of Holarctic bufonids, in coherence with fossil data (*Figure 2*, *Figure 2—figure supplement 1* and *Supplementary file 1B*).

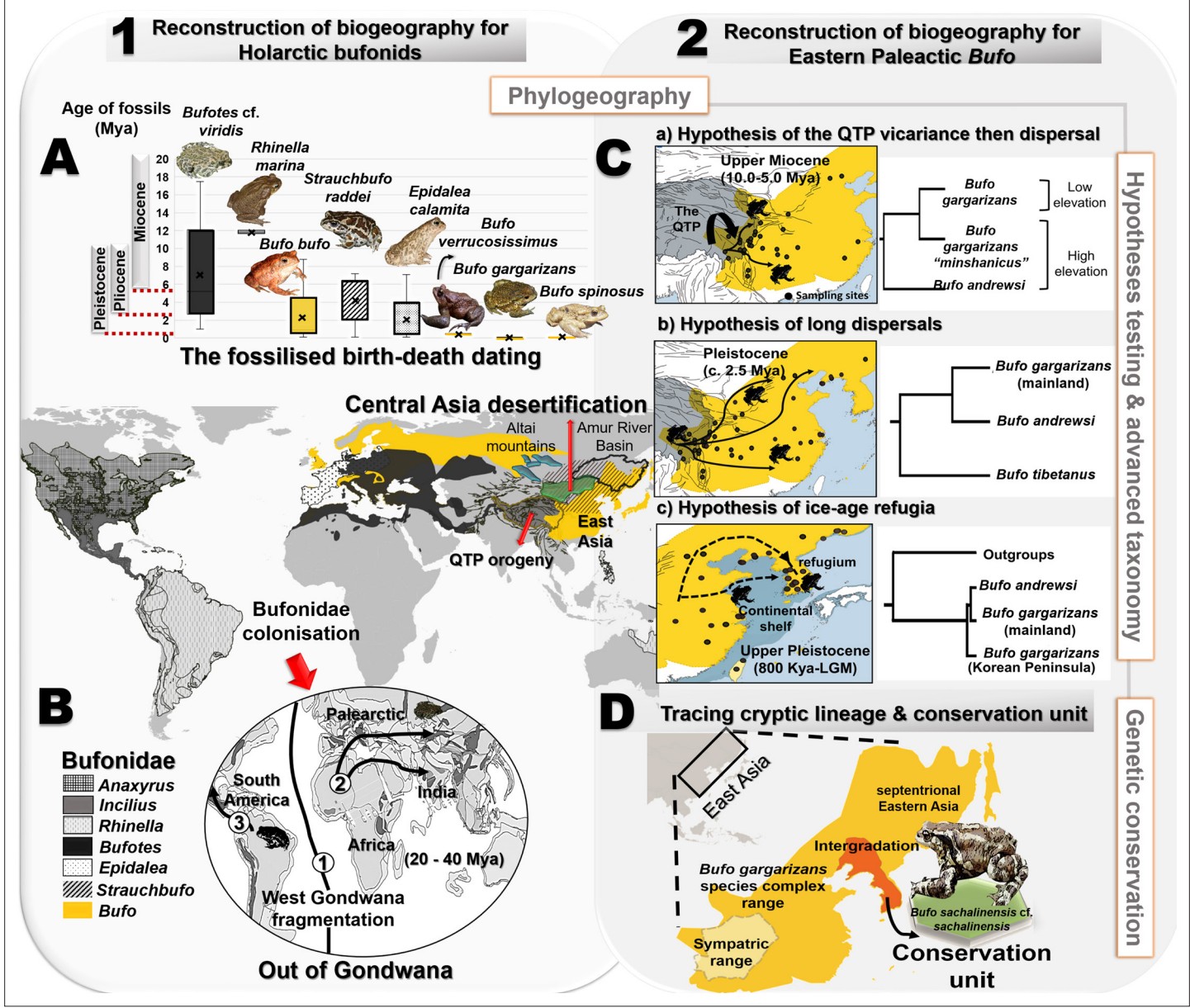

**Figure 1.** Resolution of the diversifications of bufonids in the Holarctic and the Eastern Palearctic. (**A**) Dating of the species tree of Holarctic bufonids, refined using the fossilized birth-death method. Here, the box plots represent the age range of fossils for each focal species. (**B**) Hypothesized dispersal pathways of genus *Bufo* in the Palearctic based on the Gondwana origin and central Asia desertification hypotheses. The map displays the range of the bufonids genera used in the dating analyses. (**C**) Molecular dating estimates and ancestral range reconstruction addressing three phylogeographic hypotheses derived from *Macey et al., 1998*, *Fu et al., 2005* and *Borzée et al., 2017* to elucidate the evolutionary history of East Asian *Bufo*. (**D**) Retracing the hidden diversity in the *Bufo gargarizans* species complex in East Asia and determining taxonomic and conservation units.

The online version of this article includes the following figure supplement(s) for figure 1:

**Figure supplement 1.** Chronology of species description in the East Asian *Bufo* genus and timeline of taxonomy updates.

## Fossilized birth death dating

The dated species tree using fossilized birth-death showed that following the split of West Gondwana, bufonids first diverged into American clades and African-Eurasian clades. The subsequent splits of genera of both ancestors followed a contemporary timeline, with the first divergence dating from the Early Oligocene to Middle Miocene (*Figure 2*).

In Eurasia, radiations between the Oligocene and Early Miocene resulted in the segregation of the clades of North African origin c. 26.08 Mya (95% highest posterior density [HPD 95%]/Mya for

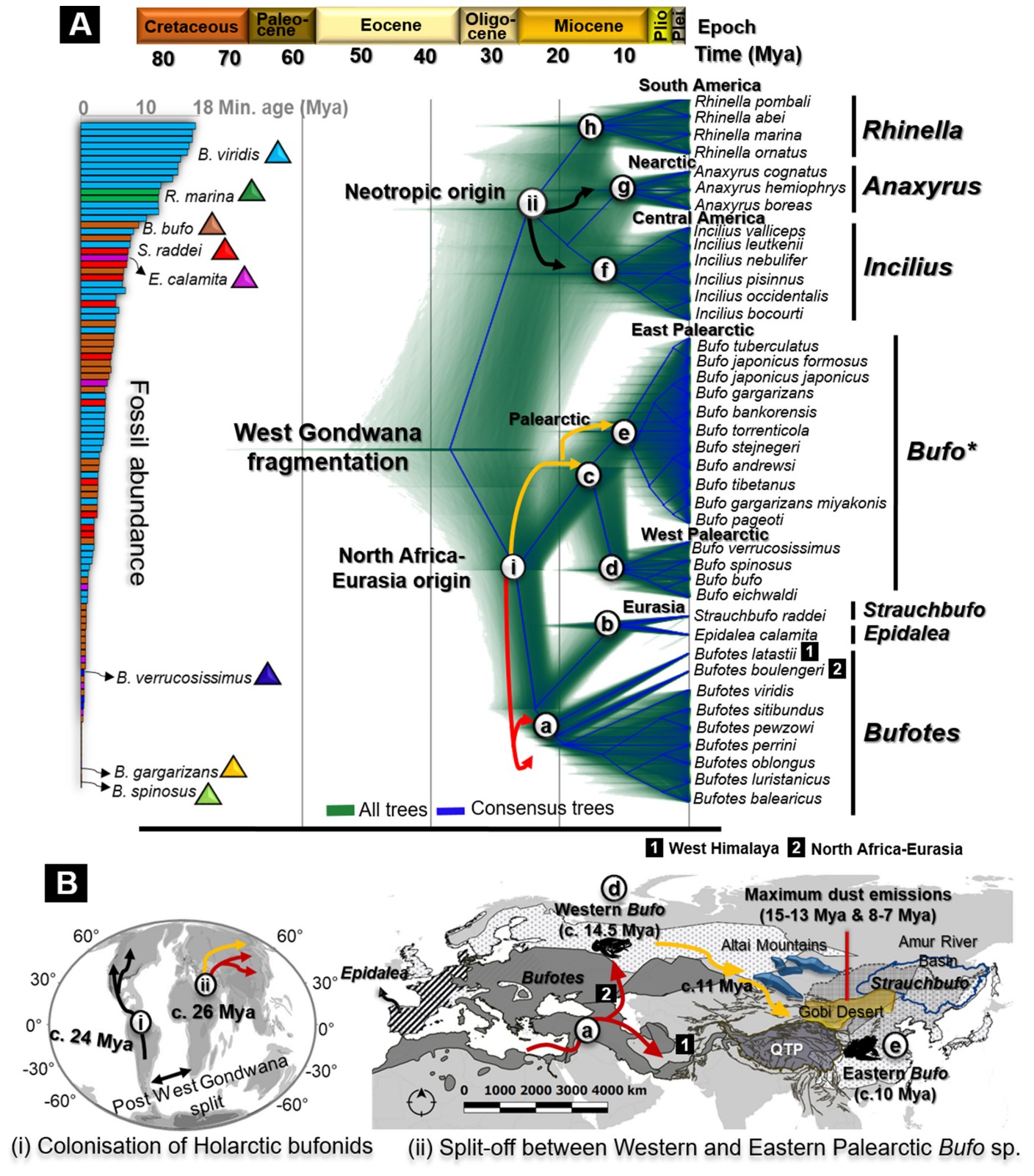

**Figure 2.** Fossils used for calibration and molecular dating of Holarctic Bufonidae. (**A**) Fossilized birth-death species trees for 39 Holarctic bufonids characterized from the unlinked multi-locus *CR-16S-ND2-CXCR4-POMC-RAG1-Rho* with an abundance of fossils representative of the six genera of Bufonidae used as source of primary calibrations. The captions (**i**) and (**ii**) on the tree are geographically explained in (**B**), representing the hypothesized dispersal pathways of Holarctic bufonids and *Bufo* spp. in the Palearctic. Similarly, the letter-coded branches of the trees are marked on the map. The

*Figure 2 continued on next page*

*Figure 2 continued*

map displays the range of bufonids genera with relevant natural features in Central Asia. Black, red, and yellow arrows in dated trees and maps indicate the dispersal pathways predicted for neotropic bufonids, *Bufotes,* and *Bufo* in the Palearctic, respectively.

The online version of this article includes the following figure supplement(s) for figure 2:

**Figure supplement 1.** Fossil abundance and distribution of Holarctic bufonids.

all dating estimates in *Table 1*; *Figure 2*). The emergence of the main extant clade was dated from the Early to Middle Miocene. The monophyletic *Bufotes* emerged c. 21.72 Mya (*Table 1*; *Figure 2*) with the North African-Eurasian group diverging from the Western Himalayan *Bufotes* c. 19.81 Mya (*Table 1*; *Figure 2*). Later, the Iberian *Epidalea* and its sister genus the Eurasian *Strauchbufo* emerged c. 10.88 Mya (*Table 1*; *Figure 2*). The Palearctic *Bufo* split into two clades, the Western and the Eastern Palearctic *Bufo* between the Early and the Middle Miocene c. 14.49 Mya (*Table 1*; *Figure 2*). The credible interval showed a considerable overlap in the timing of radiations within *Bufo* during the Middle Miocene (10.00–15.00 Mya), indicated by the isolation of the European *Bufo* clade c. 11.03 Mya (*Table 1*; *Figure 2*), and a subsequent emergence of the Asian *Bufo* clade c. 9.99 Mya (*Table 1*; *Figure 2*).

The earliest split among the American bufonids segregated the Neotropical clades predominantly during the Early Miocene c. 24.25 Mya (*Table 1*; *Figure 2*). The most basal divergence was *Rhinella* in the southern continent c. 14.16 Mya (*Table 1*; *Figure 2*), from which the younger Central American *Incilius* c. 11.90 Mya (*Table 1*; *Figure 2*), and Nearctic *Anaxyrus* may have diverged c. 9.56 Mya (*Table 1*; *Figure 2*). The alternative calibrated time tree analysis under a Yule speciation model resulted in a topology and time estimates that generally corresponded to that of the fossilized birth-death analysis (*Table 1*).

## Evolutionary diversification of Eastern Palearctic *Bufo*

Here, we focused on reconstructing the historical biogeography of Eastern Palearctic *Bufo* and resolving the taxonomic boundaries of the species complex within the genus. The vast distribution

**Table 1.** Timeframe estimate from the coalescent species tree for Holarctic bufonids following the Gondwanan origin hypothesis.

The key nodes represent the speciation events based on six internal calibrations of the minimum age range obtained from 102 fossils records under a log-normal distribution as priors. Dating analyses of the bufonids species tree shows the comparable datation estimated under a relaxed clock with the fossilized birth-death and Yule speciation models. The label for each clade is matched with the species tree in *Figure 2*.

| Clade | Key nodes | Dating analysis methods | |
|---|---|---|---|
| | | Fossilized birth-death (median [HPD 95%]/Mya) | Yule (median [HPD 95%]/Mya) |
| i | North Africa-Eurasia origin | 26.08 [22.54–32.51] | 25.88 [22.55–31.81] |
| ii | Neotropical origin | 24.25 [13.21–39.07] | 23.40 [13.10–35.61] |
| a (1) | Emergence of *Bufotes* (West Himalaya) | 21.72 [18.90–25.25] | 21.67 [19.63–27.68] |
| a (2) | Emergence of *Bufotes* (North Africa-Eurasia) | 20.42 [18.10–23.28] | 20.37 [18.79–25.02] |
| c | Emergence of Palearctic *Bufo* | 14.49 [9.76–22.70] | 14.52 [9.83–22.58] |
| d | Radiation of western Palearctic *Bufo* | 11.03 [9.14–15.42] | 11.04 [9.12–15.37] |
| e | Emergence and early radiation of eastern Palearctic *Bufo* | 9.99 [4.66–16.57] | 10.11 [4.58–6.34] |
| b | Emergence of *Epidalea* and *Strauchbufo* (Eurasia) | 10.88 [7.11–17.78] | 10.87 [7.03–17.51] |
| h | Emergence of *Rhinella* (South America) | 14.16 [11.16–22.24] | 14.00 [11.15–21.54] |
| g | Emergence of *Incilius* (Central America) | 11.90 [7.15–22.40] | 11.80 [7.16–21.52] |
| f | Emergence of *Anaxyrus* (Nearctic) | 9.56 [6.16–18.14] | 9.53 [6.13–17.58] |

and taxonomic inconsistencies in Eastern Palearctic *Bufo* warrant a careful examination of the hypotheses proposed by previous studies.

## Optimum species tree topology for Palearctic *Bufo*

The vast distribution range of the *Bufo* genus in the Eastern Palearctic resulted in cryptic diversity. Hence, we aimed to resolve the topology of Palearctic *Bufo* lineages (N species =26) following five hypotheses (*Figure 3* and *Figure 3—figure supplement 1*). We derived these hypotheses from ranges, life histories, geological events, and the likelihood of single or multiple origins. Here, our focal taxa included the Palearctic *Bufo* species and three group members of the '*Torrentophryne*' genus, a clade paraphyletic to *Bufo*, to elucidate the validity of the genus and determine its relationship with Asian *Bufo* (*Figure 3* and *Figure 3—figure supplement 1*). The nested sampling analyses supported an optimum tree topology for the Eastern Palearctic *Bufo* linked to Miocene geological events with the highest Marginal Likelihood estimation (MLE; Model C: *Supplementary file 1C*, *Figure 3*), followed in likelihood by the topology of a single origin for the East Asian *Bufo* clade (Model E: *Supplementary file 1C*, *Figure 3*). Whereas, the topologies structured by life history and morphological trait recovered the lowest likelihoods, and did not favor the monophyly of '*Torrentophryne*' (Model B: *Supplementary file 1C*, *Figure 3*). The best-fit topology of Model C supported three well-resolved monophyletic clades, a *Bufotes* (PP: 1.0; *Figure 3*), a Western Palearctic *Bufo* clade (PP: 0.90; *Figure 3*), and an Eastern Palearctic *Bufo* clade (PP: 1.0; *Figure 3*). Despite weakly recovering the East Asian mainland *Bufo* clade, the species tree strongly supported the distinction of the Japanese *Bufo* subclades (PP: 1.0; *Figure 3*). The alternative topologies from the four suboptimum models (Models A, B, D, and E) were qualified by lower MLE values than that of Model C (details in *Supplementary file 1C* and *Figure 3—figure supplement 1*).

## Phylogeny and population structure of East Asian *Bufo*

We first inferred the haplotype network of both mtDNA and nuclear data using the median-joining method. The haplotype of the concatenated mtDNA resulted in 98 haplotypes (see genetic diversity and neutrality tests in *Supplementary file 1D*; *Figure 4A*). The AMOVA based on the six monophyletic clades (N populations =8) recovered from the mtDNA phylogenetic analyses supported the population structure and showed that 55.48% of the molecular variance was attributed to differences among clades (df =5). We found 22.08% of the molecular variance among populations within clades (df =7) and 22.44% of the variance was found within individuals (df =213; *Supplementary file 1E*). The average fixation index over all the loci tested showed that $F_{SC}$ =0.50, $F_{ST}$ =0.78, and $F_{CT}$ =0.55, and that there was a negative correlation between geographical distance and genetic differentiations (N populations =13; Pearson' r=–0.059, *Figure 4—figure supplement 1*). The analysis of the nuclear data (*POMC-RAG-1-Rho*) from eight populations resulted in a single group of haplotypes (N=54) with a haplotype diversity (Hd) of 0.972 (*Figure 4B*), an average pairwise difference of 7.667 (±4.533), and a nucleotide diversity of 0.007 (±0.005). Additionally, the Mantel test for nuclear data revealed a low correlation between the geographic distance and genetic variation for the diploid populations of the East Asian *Bufo* (N population=8; Pearson's r=0.072, *Figure 4—figure supplement 1*).

To increase the phylogenetic resolution of *Bufo* across East Asia, we enlarged the sampling range to septentrional Eastern Asia where it covered the distributions of *B. sachalinensis sachalinensis* in the Amur River Basin and the *B. sachalinensis* cf. *sachalinensis* subclade restricted to the Korean Peninsula (currently '*B. gargarizans*'). We also conducted two independent molecular phylogenetic analyses using concatenated mtDNA fragments of the control region (*CR*) and NADH dehydrogenase 2 (*ND2*); and concatenated the nuDNA of the gene fragments proopiomelanocortin (*POMC*), recombination activating gene 1 (*RAG-1*) and rhodopsin (*Rho*). The phylogenetic trees derived from the mtDNA (N taxa =221) and nuDNA data (N=44) inferred six monophyletic clades, respectively. The mtDNA and nuDNA trees recovered discordant topologies, and they resulted in different composition of clades (*Figure 4C and D*). We found *B. gargarizans* from the Asian mainland to be segregated into multiple clades scattered across the nuDNA tree (see details in *Table 2*, *Figure 4*). We highlighted other discordance found between the monophyletic *B. j. formosus* and *B. j. japonicus* in the mtDNA tree (Clades 1 and 2; *Figure 4C*), in which both Japanese *B. japonicus* clustered in a clade based on the nuDNA tree (Clade 4; *Figure 4D*). In addition, nuDNA resolved *B. bankorensis* and the septentrional East Asia clades of *B. sachalinensis* as monophyletic (*Figure 4D*). Despite these inconsistencies, we found both

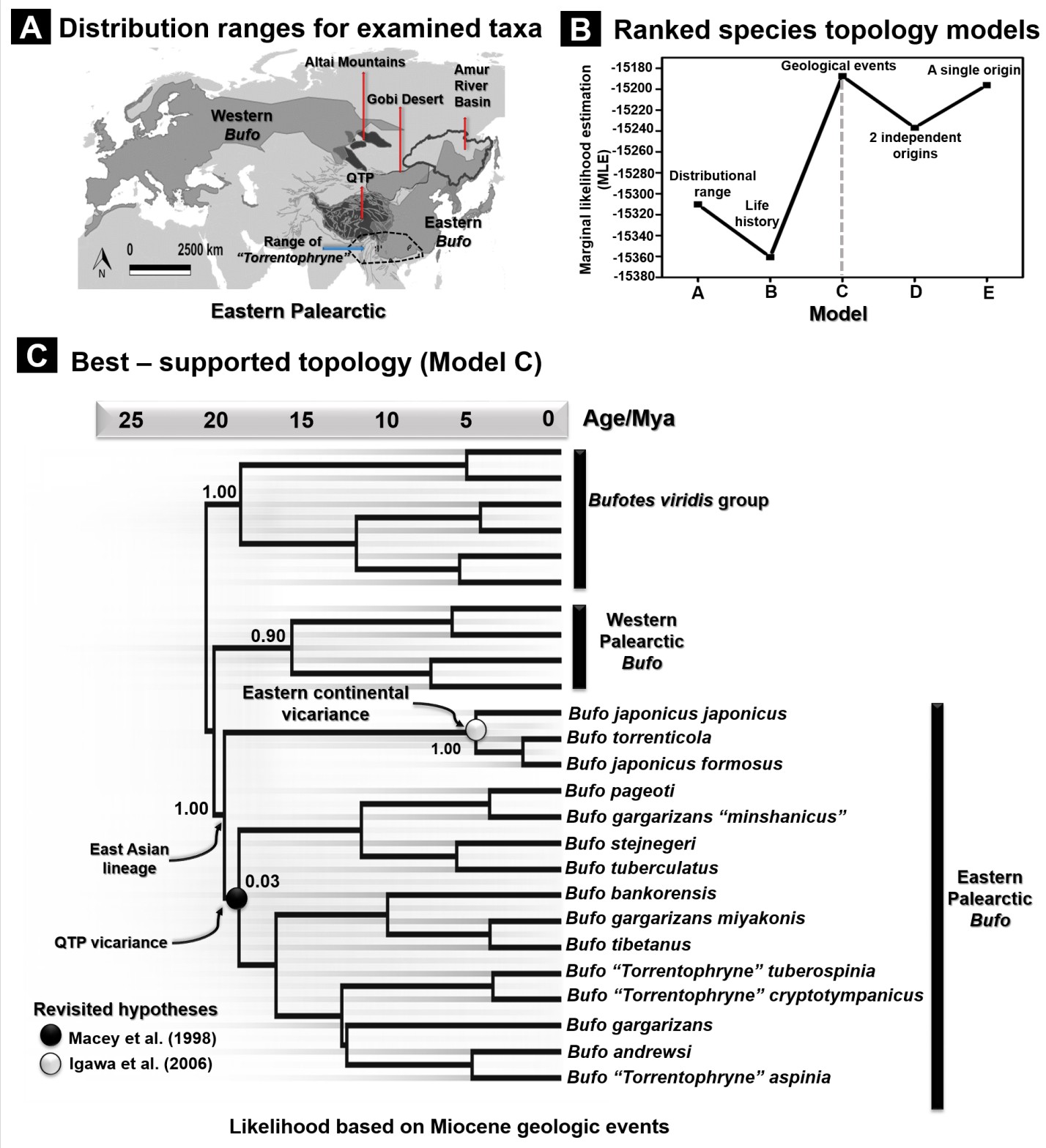

**Figure 3.** Species tree topology estimates based on hypotheses for 26 recognized Palearctic bufonids. (**A**) Geographic range of the Palearctic *Bufo* genus and related '*Torrentophryne*' species included in the reconstruction of the species tree. (**B**) Ranking of the topology models for the species tree examined using nested sampling analyses. (**C**) Best-supported topology on dated species trees reconstructed from the unlinked multilocus data (*CR-16S-ND2-CXCR4-POMC-RAG1-Rho*) under a relaxed clock and Yule prior. Each model tree (models A–E) represents the hypothesis tested for the tree

*Figure 3 continued on next page*

*Figure 3 continued*

topology with the rank of its likelihood based on the marginal likelihood estimation (MLE) values. The geology driven factor (model C) was selected to be the most accurate scenario for the species tree topology due to the highest value of MLE.

The online version of this article includes the following figure supplement(s) for figure 3:

**Figure supplement 1.** Alternative models of species tree topologies for Palearctic *Bufo*.

mtDNA and nuDNA trees to similarly manifest a segregation between the septentrional East Asian clades of *B. sachalinensis* and *B. gargarizans* (*Figure 4*).

Additionally, we reconstructed a 16S rRNA only tree to include an individual *B. gargarizans* from Vietnam. The individual did not cluster with East Asian lowlands *B. gargarizans*, rather it was nested inside a clade of high elevation-restricted species including *B. andrewsi* and *B. tibetanus* (Clade C: *Figure 4—figure supplement 2*).

## Diploid genotype clusters

The structure analysis based on the 1030 bp of multilocus *POMC-RAG-1-Rho* supported two clusters (K=2) within East Asian *Bufo*, with Mean (LnProb) equal to –827.838 and mean (similarity score) among 10 runs equal to 0.974 (*Figure 4D*). The first cluster included the populations of *Bufo* in the western, central, southeastern, and northeastern Asian mainland along with the Japanese *Bufo* (*Figure 4D*). We recorded a negligible amount of admixture between population clusters of *B. gargarizans* in the western mainland, eastern mainland, and the Japanese Archipelago (see admixture portions in *Supplementary file 1F* and *Figure 4D*). However, we found significant amounts of admixture in the central and southeastern mainland Asia (*Supplementary file 1F*, *Figure 4D*). The second cluster was restricted to *B. sachalinensis* of septentrional East Asia with significant admixture, ranging from the Korean Peninsula to the Amur River Basin and Sakhalin Island (*Supplementary file 1F*, *Figure 4D*). This cluster also included the population of *B. bankorensis* distributed in Taiwan Island (*Figure 4D*).

## The effect of introgression

The cytonuclear discrepancy (*Figure 4*) may be the result of introgression or/and incomplete lineage sorting. Thus, we further evaluated the pattern of introgression in our nuclear data (*POMC-RAG1-Rho*, N=44, nucleotide length =1030 bp) using ABBA or BABA test. To do so, we employed Patterson's D-statistic to compare the number of allelic ABBA and BABA sites. Here, the D-statistic value we obtained was equal to 1, with the ABBA-BABA pattern calculated among the sites failing to reach equal frequencies (50:50). The probability of specific sites carring the allelic patterns of ABBA or BABA was equivalent to 1, and the number of segregating sites that fit the pattern of ABBA or BABA in at least one population was equivalent to 0. The unsymmetrical frequencies violate the assumption that only incomplete lineage sorting affects the *Bufo* nuclear tree, and showed the possibility of introgression as a significant factor in shaping the nuclear genetic structure.

## Molecular dating and ancestral range

Despite topological discordance, all nuDNA and mtDNA trees consistently revealed a distinction between the clades of Eastern Central Asia and septentrional East Asia in the *B. gargarizans* complex. Thus, we also provided dating estimates using the nuDNA data set as an alternative to the mtDNA estimates (*Table 3*). In the context of dating estimates, we considered mtDNA estimates more informative than nuclear estimates based on two factors: (1) the higher number of taxa in the mtDNA tree than in the nuDNA tree resulting to a clearer phylogeographic structure with higher support values for the clades recovered and (2) the clades recovered from the mtDNA tree highly matched the best-supported topology of species tree of Eastern Palearctic *Bufo* (*Figure 2*). Overall dating for the major *B. gargarizans* clades from the nuDNA data set (especially the 95% HPD ranges) were in agreement with the mitochondrial estimates, with nuDNA ingroup nodes slightly younger than mtDNA (*Table 3*).

Here, we compared the three hypotheses related to the phylogeography of the *B. gargarizans* complex, explained by: QTP vicariance and dispersal (*Macey et al., 1998*), dominance of long dispersals (*Fu et al., 2005*), and ice-age refugia (*Borzée et al., 2017*). Our dating and ancestral range estimates supported the contribution of vicariance and dispersal to the earliest diversification of the East Asian *Bufo*. We dated the events concerning the basal clade of Eastern Asian *Bufo* from the Early

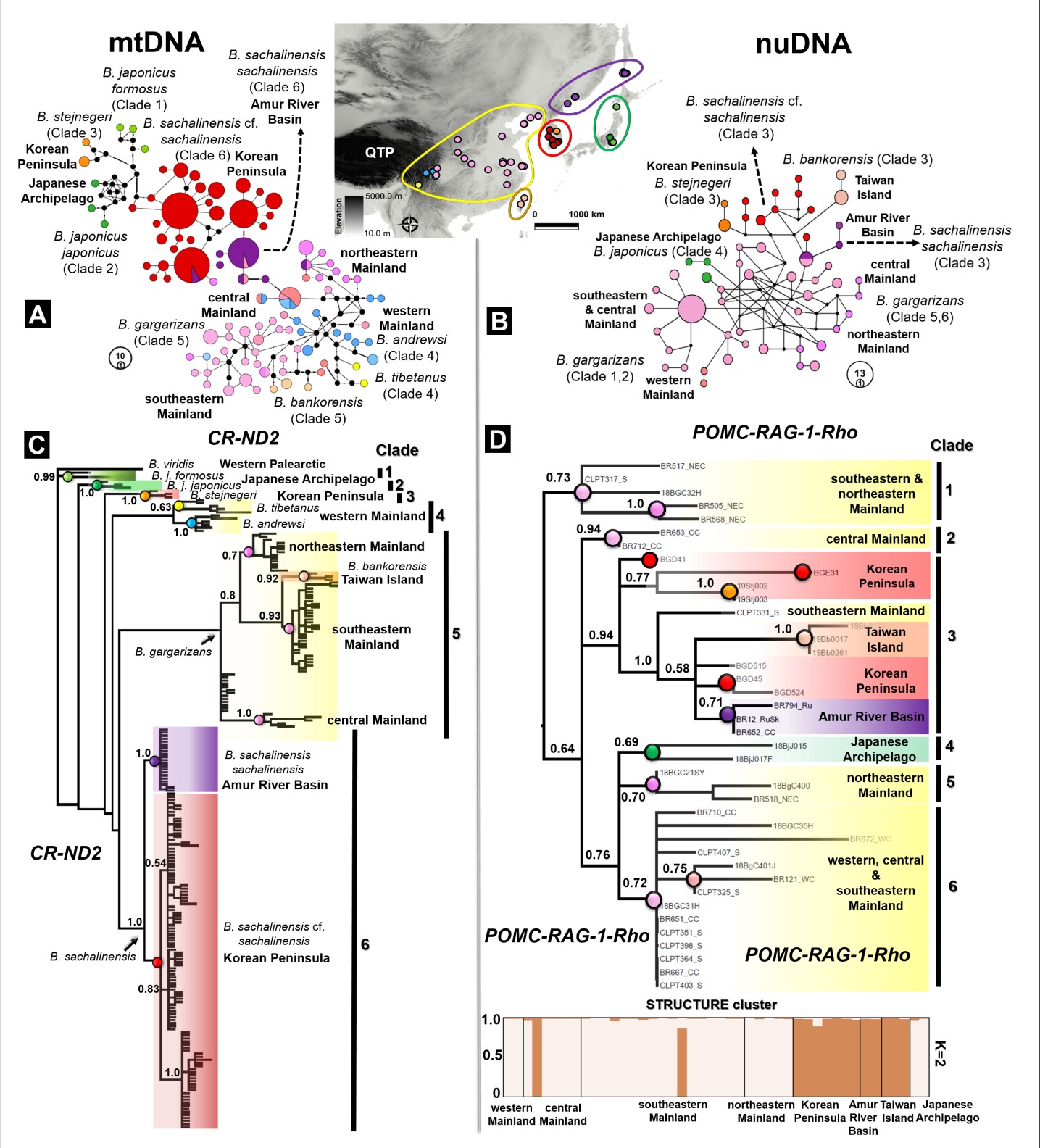

**Figure 4.** Discordance in phylogenetic and haplotype relationships of the *Bufo* genus in East Asia inferred from mtDNA and nuclear protein coding. (**A**) Median joining network based on mtDNA *CR-ND2* (894 bp) obtained from 221 individuals. (**B**) Phylogenetic relationships inferred from the same data set of concatenated mtDNA *CR-ND2*. (**C**) Haplotype relationship of 44 individuals of East Asian *Bufo* derived from diploid data (nuclear *POMC-RAG-1-Rho*; 1030 bp). The discordance in topology between the mitochondrial and nuclear trees is detailed in *Table 2*. (**D**) Phylogenetic relationship and

*Figure 4 continued on next page*

*Figure 4 continued*

population clustering inferred from SNP of the nuclear protein coding analyzed in STRUCTURE and CLUMPP (see **Supplementary file 1F** for details about the populations). The color code used in the map of East Asia matches with the colors coded for each particular clade and haplotype group.

The online version of this article includes the following figure supplement(s) for figure 4:

**Figure supplement 1.** The tests of isolation by distance on *Bufo* distributed in the Eastern Asia.

**Figure supplement 2.** The phylogenetic relationship between Bufonidae with emphasis on East Asian *Bufo* lineage inferred on partial 16S rRNA gene fragment.

to Late Miocene c. 14.20 Mya (see HPD 95% in **Table 3**; **Figure 5**). The events were subsequently followed by the emergence and isolation of the Japanese *Bufo* from the Eastern Asian lineage in Late Miocene (**Figure 5A1**), the Japanese *Bufo* group then splitting into two distinct species, *B. j. formosus* and *B. j. japonicus*, c. 8.23 Mya (**Table 3**, **Figure 5**). These events were followed by the radiations within the East Asian *Bufo* lineage resulting in the independent divergence of *B. stejnegeri* in the Korean Peninsula, c. 8.90 Mya (**Table 3**, **Figure 5**) and a split between the high altitudes East Asian clades: *B. tibetanus, B. andrewsi,* and *B. gargarizans* c. 11.32 Mya (**Table 3**, **Figure 5**). These East Asian clades are present in the areas of the QTP and likely to have dispersed from high elevation areas (**Figure 5A2**), resulting in the divergence between *B. tibetanus* and *B. andrewsi,* dated between the Pliocene and the Late Miocene c. 5.24 Mya (**Table 3**, **Figure 5**).

We found the combined effects of long dispersal and ice-age refugia to contribute equally to the radiation of the *B. gargarizans* complex in the Plio-Pleistocene, although the trade of the species over the last millennium has muddled the genetic signature (**Figure 5A3**). The deepest split within the *B. gargarizans* complex is estimated to have occurred around the same timeframe and at high elevation, delineating the *B. minshanicus* clade (*B. gargarizans* subspecies) in the southwestern to central mainland Asia (i.e., Sichuan and Shaanxi) c. 5.27 Mya (**Table 3**; **Figure 5**). Later, *B. gargarizans* may have dispersed to lower latitudes with the splitting from the monophyletic clade of *B. gargarizans*

**Table 2.** Comparison of the cladistic characteristics between the mitochondrial and nuclear trees of East Asian *Bufo*.
The comparison between concatenated mtDNA and nuDNA trees demonstrated the discordances between the recovered clades of *Bufo gargarizans* distributed in East Asia and showed identical patterns of divergence in the septentrional East Asian *Bufo sachalinensis* clades. The number of clades in the description is matching the phylogenetic trees in **Figure 4**.

| Distribution range | Clade | Description of cladistics | |
| --- | --- | --- | --- |
| | | Concatenated mtDNA (*CR-ND2*) | Concatenated nuDNA (*POMC-RAG1-Rho*) |
| Japanese Archipelago | *Bufo japonicus formosus* | Monophyletic (Clade 1) | |
| Japanese Archipelago | *Bufo japonicus japonicus* | Monophyletic (Clade 2) | Grouped together in a clade of Japanese *Bufo* |
| Korean Peninsula | *Bufo stejnegeri* | Monophyletic (Clade 3) | *B. stejnegeri* were grouped with Korean *B. gargarizans* in a nested clade of southeastern mainland and septentrional East Asian *B. gargarizans* (Clade 3) |
| Northeastern Mainland Asia | *Bufo gargarizans* | Monophyletic (Clade) | Contained multiple clades of *B. gargarizans* restricted to the southeastern and northeastern Asian mainland |
| Eastern Mainland | *Bufo gargarizans* | Polyphyletic with a clade of *B. bankorensis* distributed in Taiwan Island (Clade 5) | Formed multiple clades across Eastern Asian *Bufo* lineages. A clade grouped with northeastern *B. gargarizans*. Another clade is polyphyletic with of *B. bankorensis* of Taiwan Island and *B. gargarizans* distributed in septentrional East Asia (Clade 3). |
| Taiwan Island | *Bufo bankorensis* | | |
| Central Mainland | *Bufo gargarizans* | | Formed two distinctive clades:<br>1. A clade that restricted to western Mainland (Clade 2).<br>2. Belonged to monophyletic groups of southeastern, western, and northeastern Mainland *B. gargarizans* |
| Septentrional East Asia (Korean Peninsula) | *Bufo sachalinensis* cf. *sachalinensis* | Monophyletic (Clade 6) | Monophyletic (Clade 3) |
| Septentrional East Asia (Amur River Basin) | *Bufo sachalinensis sachalinensis* | Monophyletic (Clade 6) | Monophyletic (Clade 3) |

**Table 3.** Molecular dating analyses for the East Asian *Bufo*.
This analysis is based on linked mtDNA genes (*CR-ND2*) and unlinked multiple nuclear protein-coding genes (*POMC-RAG-1-Rho*) under a relaxed molecular clock with different tree priors. The node numbers are matching the clades in the dated phylogenetic tree and hypothesized dispersal pathways in *Figure 5*.

| Key events (node number) | mtDNA (CR-ND2) Relaxed molecular clock | | | nuDNA (POMC-RAG-1-Rho) Strict molecular clock |
| --- | --- | --- | --- | --- |
| | Yule prior (median [HPD 95%]/Mya) | Birth-death prior (median [HPD 95%]/Mya) | Mean (median [HPD 95%]/Mya) | Birth-death prior (median [HPD 95%]/Mya) |
| Root age of East Asian *Bufo* | 10.47 [7.88–13.40] | 17.93 [11.81–26.63] | 14.20 [9.46–20.02] | 10.12 [6.60–12.77] |
| Emergence of Japanese *Bufo* (A1) | 7.75 [5.98–9.52] | 8.70 [6.97–10.44] | 8.23 [6.48–9.98] | 7.64 [4.99–8.92] |
| Crown clade of *B. j. formosus* | 3.71 [1.71–5.92] | 4.15 [2.38–6.13] | 3.93 [2.05–6.03] | – |
| Crown clade of *B. j. japonicus* | 2.24 [0.72–4.33] | 1.58 [0.57–2.84] | 1.91 [0.65–3.59] | – |
| Crown clade of *B. stejnegeri* | 3.32 [1.54–5.24] | 2.22 [0.72–4.12] | 2.77 [1.13–4.68] | 4.09 [1.46–5.43] |
| MRCA of East Asian mainland *Bufo* (A2) | 8.38 [6.13–10.88] | 14.25 [9.67–19.44] | 11.32 [7.90–15.16] | – |
| Crown clade of *B. tibetanus* – *B. andrewsi* | 5.21 [3.25–7.45] | 5.26 [3.01–7.66] | 5.24 [3.13–7.55] | – |
| Stem node of *B. gargarizans* complex (A3) | 6.85 [4.70–9.32] | 11.25 [7.28–16.08] | 9.04 [5.99–12.70] | – |
| Stem clade of Chinese mainland of *B. gargarizans* | 5.19 [3.16–6.72] | 5.34 [3.45–7.83] | 5.27 [3.31–7.28] | – |
| Segregation between *B. gargarizans* inhabiting high (*B. minshanicus*) and low elevation (*B. g. gargarizans*) | 4.70 [3.23–6.53] | 2.68 [1.82–3.26] | 3.69 [2.53–4.90] | 3.77 [2.39–3.83] |
| Crown clade of *B. andrewsi* and *B. gargarizans* inhabiting high elevated range (*B. minshanicus*) | 2.89 [2.18–3.54] | 2.55 [1.86–3.28] | 2.72 [2.02–3.41] | 2.79 [1.18–3.10] |
| Crown clade of *B. g. gargarizans* in the southeastern Mainland | 3.14 [1.92–4.49] | 2.56 [1.48–3.86] | 2.85 [1.70–4.18] | 3.05 [1.04–3.45] |
| Nested clades of *B. gargarizans popei* and *B. g. gargarizans* in the Central, southeastern and northeastern Mainland | 2.21 [1.37–3.20] | 1.28 [0.71–1.95] | 1.85 [1.04–2.58] | 3.65 [1.18–5.28] |
| Crown clade of *B. bankorensis* | 1.34 [0.74–1.94] | 1.28 [0.72–1.84] | 1.31 [0.73–1.89] | 1.80 [0.50–1.80] |
| Stem clade of septentrional East Asian *B. sachalinensis* | 1.95 [1.55–2.34] | 1.81 [1.38–2.28] | 1.88 [1.47–2.31] | 2.21 [0.93–2.16] |
| Crown clade of Korean *B. sachalinensis* cf. *sachalinensis* | 1.58 [1.18–1.93] | 1.22 [0.82–1.61] | 1.40 [1.00–1.77] | – |
| Crown clade of Russian *B. sachalinensis sachalinensis* | 1.06 [0.62–1.53] | 0.46 [0.19–0.79] | 0.76 [0.41–1.16] | – |

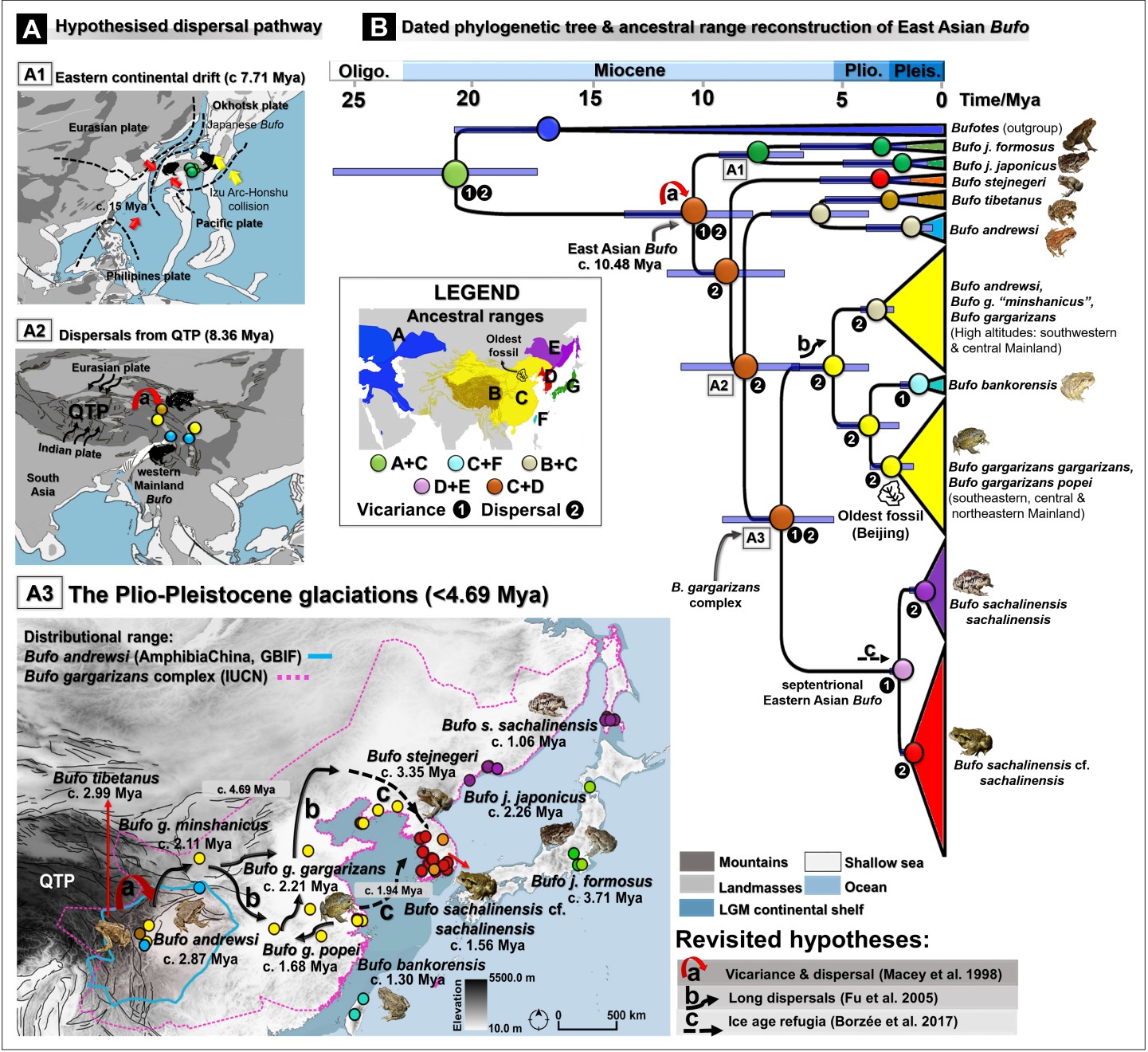

**Figure 5.** Dated phylogeny and biogeography patterns of *Bufo* in the Eastern Palearctic. (**A**) Hypothesized dispersal pathways for *Bufo* in the Eastern Palearctic. (**B**) Dated Maximum Clade Credibility (MCC) tree, ancestral ranges reconstruction, and colonization history for Palearctic bufonids inferred from the linked mtDNA *CR-ND2* (N individuals =132). The dispersal pathways illustrated indicate two Miocene vicariance events for the MRCA of East Asian *Bufo* (**A1–A2**) and the routes hypothesized to have been followed by the *B. gargarizans* complex for dispersion through the Asian mainland (**A3**). The Miocene tectonic plates models and the Tibetan-Himalayan mapping were both retrieved from established data sets (refer to **Supplementary file 1O**) and projected to this paleomap.

*gargarizans* ranging from southeastern to northeastern mainland Asia (i.e., Shanghai, Zhejiang, Hubei, Jinan, and Shenyang) c. 2.85 Mya (**Table 3**; **Figure 5**). A widely dispersed *B. gargarizans* clade further diverged in the central, southeastern, and northeastern mainland Asia (i.e., Sichuan, Shaanxi, Hubei, and Jiangsu, Dalian) c. 1.85 Mya (**Table 3**; **Figure 5**), nested within the *B. g. popei* clade (**Figure 5**). The *B. bankorensis* clade is a more recent divergence c. 1.31 Mya (**Table 3**; **Figure 5**), sharing a common ancestor with the southeastern *B. g. gargarizans* (**Table 3**, **Supplementary file 1G**, **Figure 5**).

We estimate the emergence of *B. j. formosus* in Japan to have occurred between the Early Pleistocene and the Late Miocene c.3.93 Mya (**Table 3**, **Figure 5**), followed by a recent divergence of *B. j. japonicus* between the Pleistocene and the Pliocene c. 1.91 Mya (**Table 3**, **Figure 5**). In Northeast Asia, the *B. stejnegeri* clade may have independently emerged at the Plio-Pleistocene boundary c. 2.77 Mya (**Table 3**, **Figure 5**). Finally, the *B. gargarizans* clade dispersed and diverged eastward of the Yellow Sea during the Pleistocene c. 1.88 Mya (**Table 3**, **Figure 5**), established a population on the Korean Peninsula c. 1.40 Mya (**Table 3**, **Figure 5**) and expanded further to the Amur River Basin c. 0.76 Mya (**Table 3**, **Figure 5**).

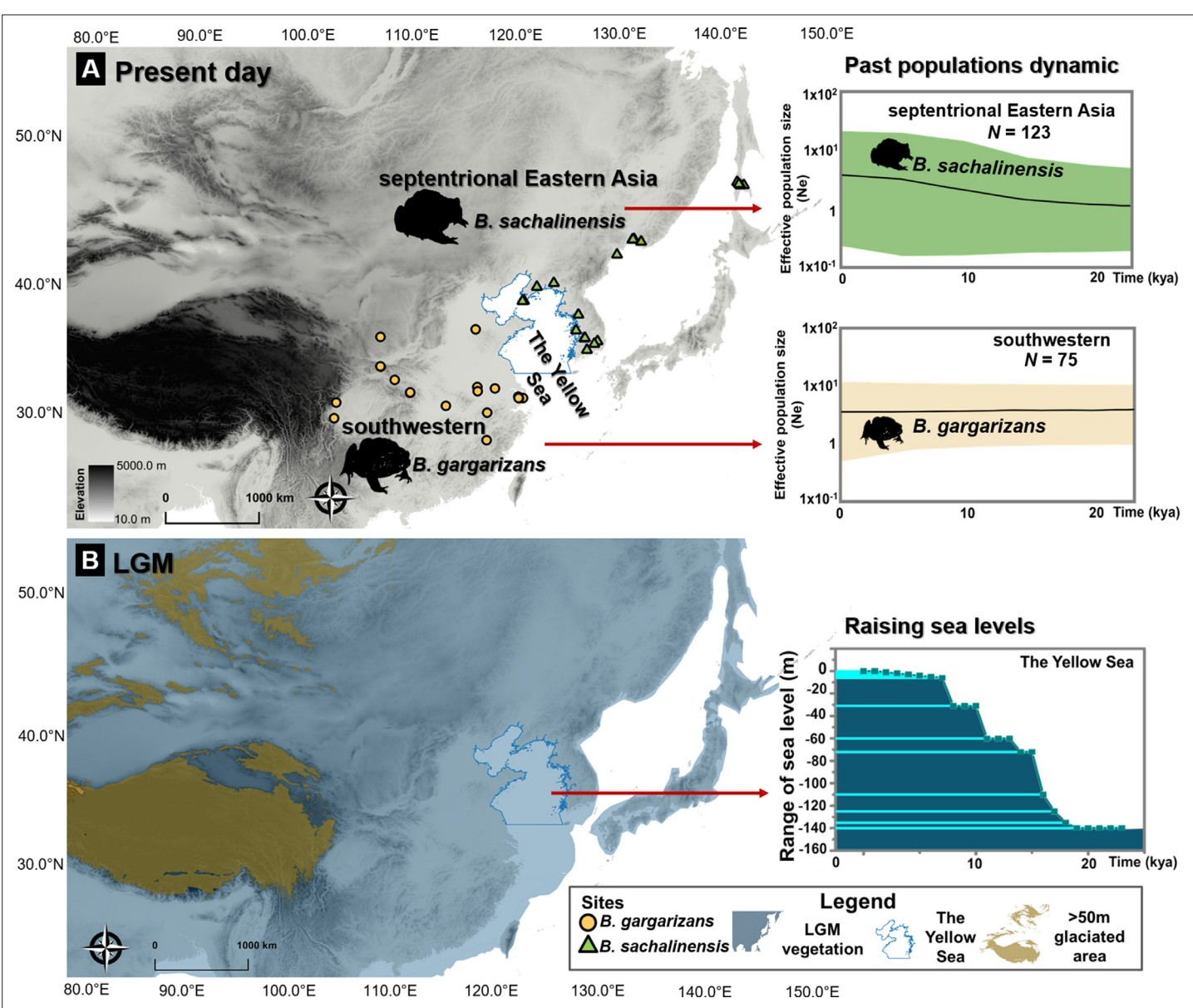

**Figure 6.** Ice age refugia hypothesis for two clades of *Bufo* in East Asia. (**A**) Past population dynamics of *Bufo sachalinensis* show an increase in effective population size (Ne). (**B**) Variability in past sea levels of the Yellow Sea since the Last Glacial Maxima (LGM). The maps represent present-day distribution of Eastern Asia *Bufo* and the LGM condition in East Asia with the projection of continental shelves during the ice ages (refer to **Supplementary file 1O**). The variation of the Yellow Sea level during LGM to present day was extracted from literature (**Li et al., 2016**). All maps were produced using QGIS v10.1 (ESRI, CA, USA).

## LGM population expansion

We tested the ice-age refugia hypothesis to infer the impact of past Yellow Sea fluctuations on the East septentrional *Bufo* clades. We demonstrated that the rise in the past Yellow Sea level resulted in population expansions for the Korean and Russian *Bufo* clades, as showed by significantly negative values of Tajima's *D* and Fu's *Fs* (refer to Clade 6; *Supplementary file 1D*). The results were consistent with the Bayesian Skyline Plot, which indicated a recent population expansion in the Amur River Basin clade c. 0.48 Mya (effective population size trajectory (mean [HPD]/Ne) 1.18 [0.30–2.94]) until present (Ne =3.73 [0.23–20.64]) with a mean likelihood: –1930.53 [HPD: –1942.89, –1918.43], *Figure 6*. In comparison, the populations of the *Bufo* distributed on the southwestern margin of Yellow Sea consistently declined since the Late Pleistocene c. 0.55 Mya (Ne =5.34 [13.46–1.124]) until present (Ne =3.43 [0.48–11.84]) with a mean of likelihood: –2954.63 [HPD: –2969.60, –2941.08], *Figure 6*.

## Ecological niche modeling

To clarify the divergence in ecological requirements between *B. gargarizans* and *B. sachalinensis*, we examined the niche overlap between the two clades (*Figure 7*). The habitat suitability model for *B. gargarizans* (*Figure 7A*) was characterized by an AUC of 0.9239±0.0185 and a TSS of 0.6741±0.0408, while the model for the *B. sachalinensis* (*Figure 7B*) was characterized by an AUC of 0.9632±0.0043 and a TSS of 0.8723±0.0115. The 'I' niche overlap statistic between the two models was 0.4198, while the 'D' statistic was 0.1566. These values were significantly lower than the average values of the null distribution, with the mean of 'I' in the null distribution being 0.9788 (p<0.0001) and the mean of 'D' in the null distribution being 0.8546 (p<0.0001). Here, our result showed the overlap to be significantly less than expected, and therefore supporting the segregation in environmental requirements between the two East Asian *Bufo* clades.

## Population migration in septentrional East Asia

We then examined the impact of gene flows and migration on local adaptation of the focal *B. sachalinensis* clades. Along the latitudinal gradient, our results suggested a weak gene flow outwards from the Amur River Basin and *B. s. sachalinensis*, in comparison with the symmetrical gene flow between the subspecies of *B. sachalinensis* restricted to the Korean Peninsula and *B. gargarizans* (*Figure 8*). This limited gene flow did not hinder the local adaptation of *B. sachalinensis* clade distributed at northern latitudes.

In particular, the migration trajectory obtained from the unlinked multi-locus data showed a lack of gene flow with a symmetrical pattern of gene exchange between the populations of *Bufo* distributed on the northeastern mainland and populations distributed on the Korean Peninsula (refer to migration rates and theta (Θ) estimates: *Supplementary file 1H*, *Figure 8*). We demonstrated a comparable and moderate gene flow when focused on the average population migration rate (M) of haploids (2 N m) and diploids (4 N m) of the two populations. The gene flow rate from the northeastern mainland toward the Korean Peninsula ($M_1$–$M_3$; *Figure 8*) was 0.106, and in the opposite direction ($M_3$–$M_1$; *Figure 8*) was 0.214 (*Supplementary file 1H*). Contrary to the symmetrical migration patterns between the northeastern mainland and Korean Peninsula, we estimated an asymmetrical migration pattern toward septentrional Eastern Asian *B. s. sachalinensis* in the Amur River Basin (*Supplementary file 1H*, *Figure 8*). We detected a higher rate of gene flow into the Amur River Basin, transferred from the eastern mainland and from the Korean Peninsula (*Supplementary file 1H*, *Figure 8*). However, we found a lower rate of gene flow from the Amur River Basin toward the northeastern mainland and the Korean Peninsula (*Supplementary file 1H*, *Figure 8*).

Additionally, the migration pattern between the two subpopulations of *B. stejnegeri* in Korea was also symmetrical, with a negligibly low rate of gene flow from the northern toward the southern population, and vice-versa (*Supplementary file 1H*, *Figure 8*).

## Species boundaries and taxonomy updates

Here, we provided support to delimit the cryptic *B. s. sachalinensis* distributed in septentrional Eastern Asia (Amur River Basin). The path sampling and Bayes factor analyses generally supported lineage-splitting as the best speciation pattern as opposed to the lumping of the clades of East Asian *Bufo* (*Supplementary file 1I*, *Figure 9*). Out of the eight species delimitation models tested (models A–H; *Supplementary file 1I*, *Figure 9*), Bayes factors determined Model A to be the most favorable

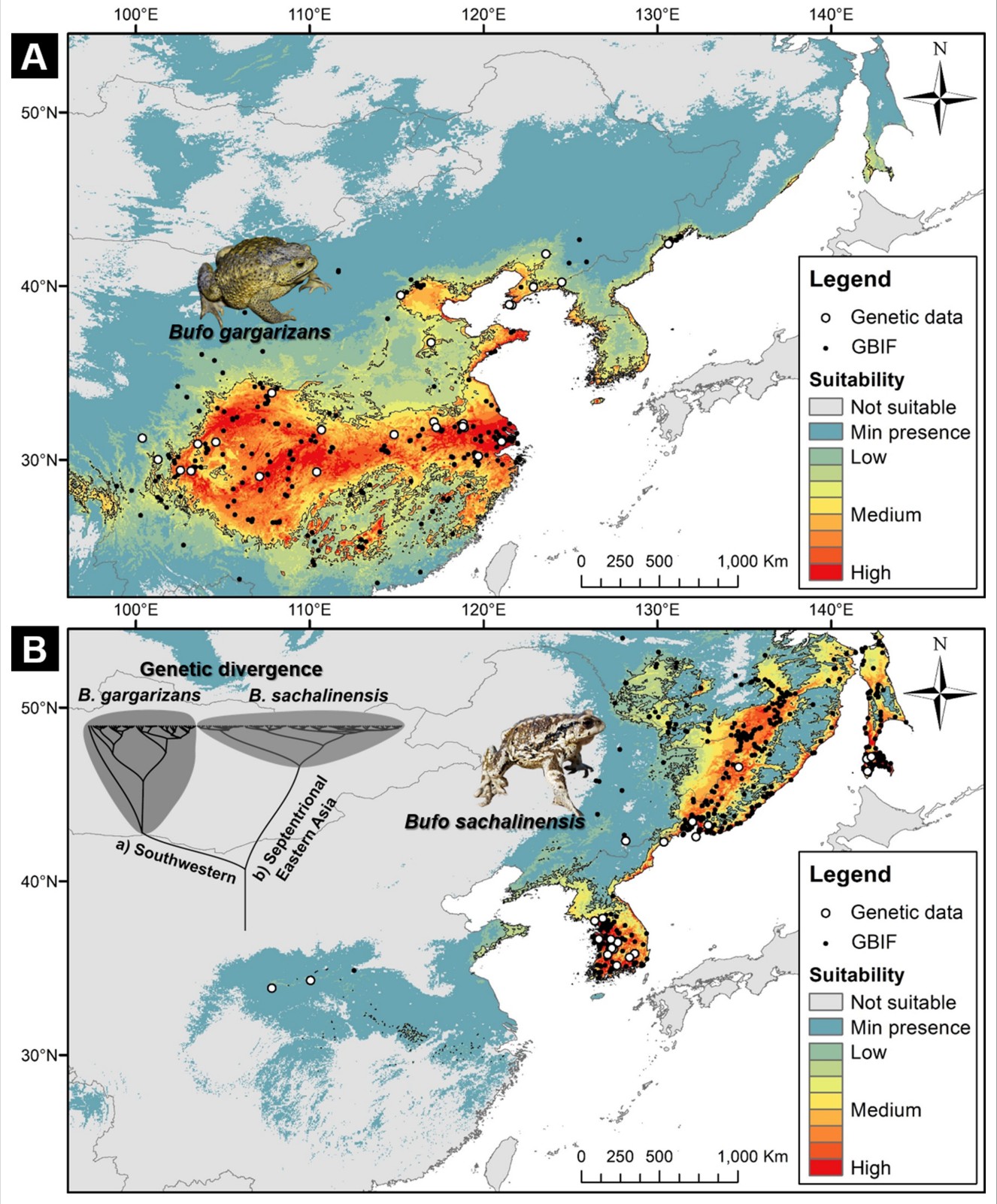

**Figure 7.** Niche suitability models for two segregated clades of East Asian *Bufo*. (**A**) The niche suitability of the southwestern clade of *B. gargarizans*. (**B**) The divergence in niche suitability of the septentrional East Asian clade of *B. sachalinensis*. The colors represent the climate suitability of the populations with the red area indicating the highest suitability, the gray area represents unsuitable habitats, and black lines represent the maximum

*Figure 7 continued on next page*

*Figure 7 continued*
sensitivity plus specificity threshold. The white dots represent the occurrence based on the phylogenetic relationship data, and the black dots represent the occurrence recorded from the Global Biodiversity Information Facility (GBIF).

The online version of this article includes the following figure supplement(s) for figure 7:

**Figure supplement 1.** Niche segregation between *Bufo sachalinensis* and *Bufo gargarizans*.

alternative, receiving the highest support out of all scenarios determined by the path sampling analyses (***Supplementary file 1I***, ***Figure 9***). Our model supported the split of the East Asian *Bufo* genus into seven independent taxonomic units (Model A; MLE: −502.801, Bayes factor: 3.198; ***Supplementary file 1I***, ***Figure 9***). These correspond to *B. j. japonicus*, *B. j. formosus*, *B. andrewsi*, *B. gargarizans*, *B. stejnegeri*, *B. bankorensis,* and the clade we refer to as *B. sachalinensis* (see justification below). Model A also supported a species-level boundary between the two subspecies of *B. japonicus*: *B. j. japonicus* and *B. j. formosus*. The species delimitation model suggested the taxonomic merging of *B. sachalinensis* cf. *sachalinensis* clade in the Korean Peninsula with the *B. s. sachalinensis* clade in the Amur River Basin (***Supplementary file 1I***, ***Figure 9***). Moreover, Model A validated the split between

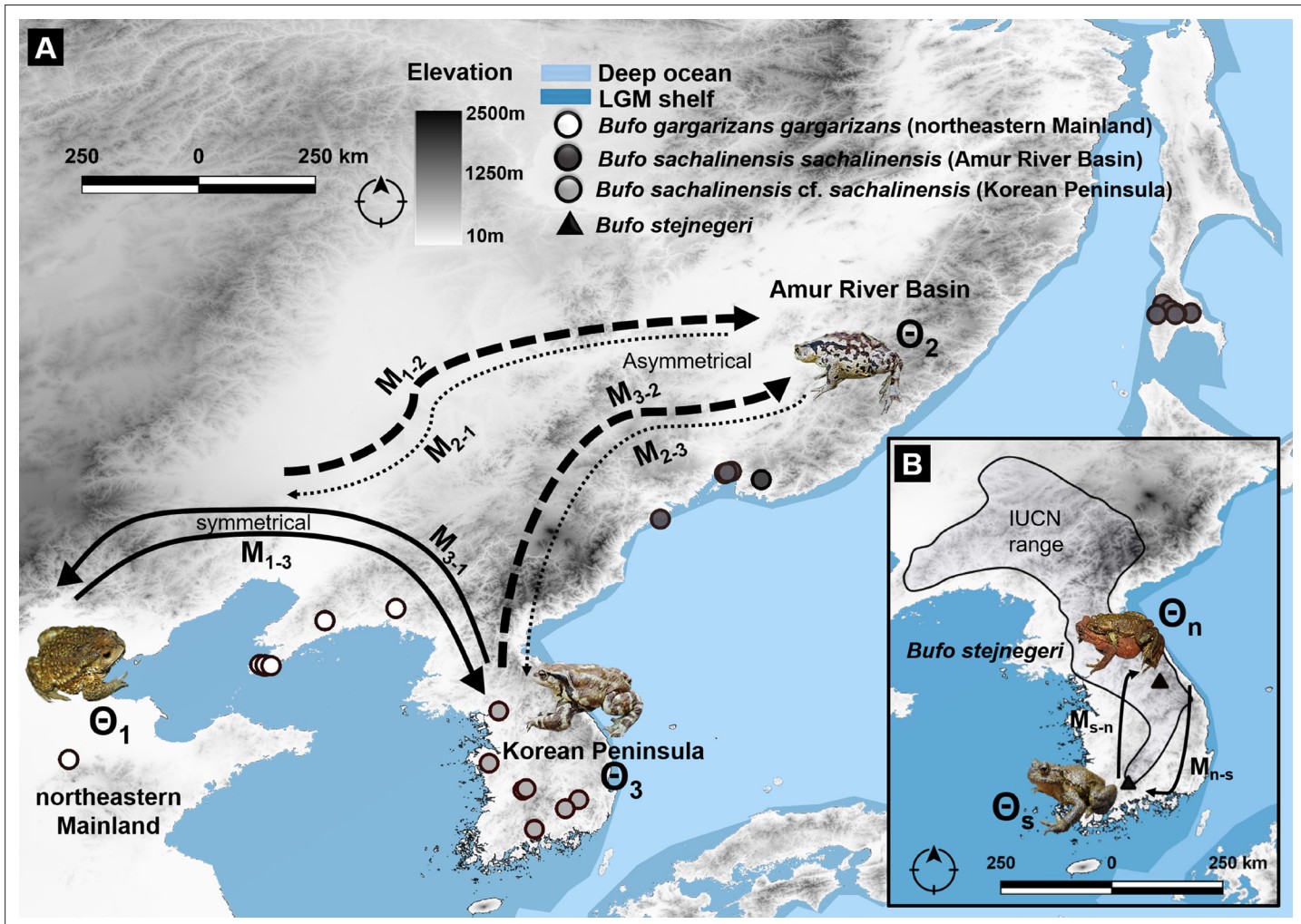

**Figure 8.** Migration trajectory estimated using MIGRATE-N among three northern latitudinal populations of *Bufo* characterized by mtDNA and nuDNA data. (**A**) Predicted migration pattern of *Bufo* in the Northeast Asia, the Korean Peninsula, and the Amur River Basin. We detected an asymmetric flow from and toward the Amur River Basin, indicating a weak gene flow from *B. s. sachalinensis* and reflecting a potential local adaptation to the climatic condition of the area. (**B**) The symmetrical pattern of migration between the north and south populations of *B. stejnegeri* in the Republic of Korea. Bold arrows indicate higher rate and thin-dashed arrows indicate lower rate in one asymmetric migration between two regions.

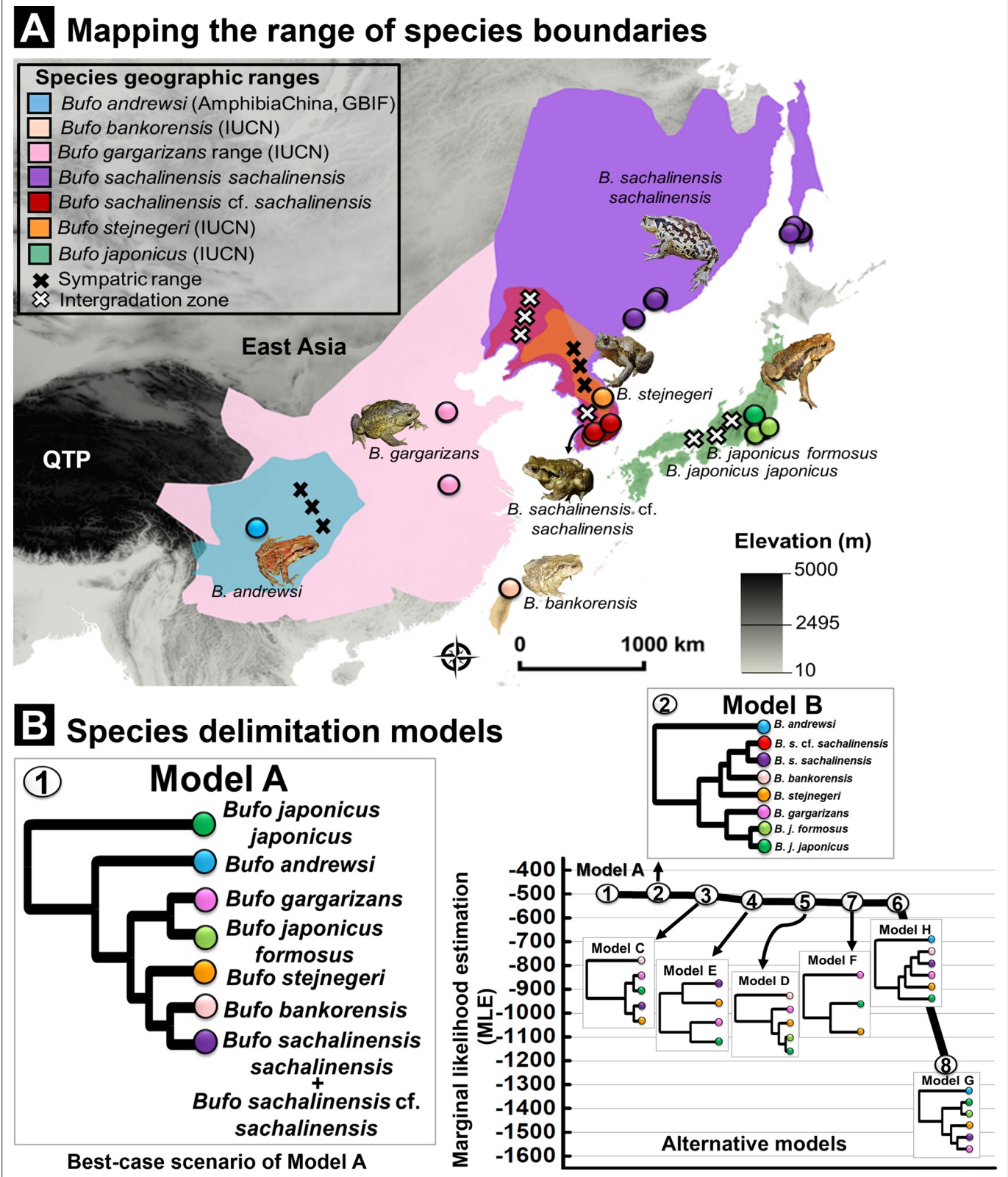

**Figure 9.** Species delimitation test using coalescent BFD approach inferred from nuDNA data (*RAG-1- POMC-Rho*). (**A**) Geographic range for Eastern Asia *Bufo* based on best-supported species delimitation model. The range of each valid species is colored following the species color codes in the map. The marking for sympatric and intergradation range (the overlapped range between two subspecies in a broad species complex range) are indicated in the legend. (**B**) Topology of species delimitation Model A, followed by Model B as the best-supported scenarios among the eight alternative models

*Figure 9 continued on next page*

*Figure 9 continued*

tested. The ranking is supported through the highest Marginal likelihood estimate (MLE) and positive Bayes factor values. The comparison of all alternative models is explained in *Supplementary file 1I*.

the clades distributed in septentrional East Asia from *B. g. gargarizans* clades in East Asia (Model A; *Supplementary file 1I*, *Figure 9*).

Following multiple calls for a taxonomic revision and the various species descriptions and synonimization (*Matsui, 1986*; *Igawa et al., 2006*; *Borzée et al., 2017*; *Lee et al., 2021*), we suggest the following updates to the taxonomy. It is to be noted that we do not describe any new species as a long list of valid former names is available (*Supplementary file 1I*, *Figure 1—figure supplement 1*). We provided the taxonomy updates on the basis of species delimitation analysis (*Figure 9*), and the following recommendations were corroborated by the allopatric distribution of the delineated clades (*Figure 9*) and the differentiation in niche requirements (*Figure 8*). Thus, we presented the most robust taxonomic framework applied to this group since the description of *B. gargarizans*.

Here, we refer to *Bufo gargarizans Cantor, 1842* (see *Cantor, 1842*) type locality in *Supplementary file 1A* and *Figure 1—figure supplement 1* as *B. gargarizans gargarizans* (the Zhoushan toad based on the type locality) following the literature (*Dubois and Bour, 2010*). The subspecies is currently geographically widespread, a consequence of its synonimization with several clades of taxonomically unstable *Bufo* in the East Asian mainland, including the intergraded subspecies *B. gargarizans popei Matsui, 1986*, two high elevation-restricted *Bufo* species described under the epithets of *B. andrewsi Schmidt, 1925*, and *B. minshanicus Stejneger, 1926* (see type localities and ranges in *Supplementary file 1A*, *Figure 1—figure supplement 1*). Here, we provide evidence for the independent evolution of *B. andrewsi* (*Figure 9*) and therefore the validity of *B. andrewsi* at the species level, against the previous synonimization (*Fu et al., 2005*). The distinction between *B. bankorensis* and *B. gargarizans* (*Figure 8*) is supported by the polyphyletic structure in their mtDNA lineage (*Figures 4B and 5*), further reinforced by the independent evolutionary lineage as shown by the multilocus BFD tree (*Figure 9*). Hence, we support the species status of *B. bankorensis* and reject their synonimization (*Liu et al., 2000*). The species was described under the epithet *B. bankorensis Barbour, 1908* (*Supplementary file 1A*, *Figure 1—figure supplement 1*).

Most notably, we propose to redefine the *Bufo* clades in septentrional East Asia (*Figure 9*), which includes the clades referred to as *B. sachalinensis* cf. *sachalinensis* and *B. s. sachalinensis* in the analysis section. We therefore recommended the elevation of the septentrional East Asia clade currently named *B. gargarizans* to the species level, under the taxonomic epithet *Bufo sachalinensis* (Life Science identifier LSID urn:lsid:zoobank.org:pub:14938673-9CEB-4325-BDBC-83A0AC85B8A7, accessible at http://zoobank.org/References/14938673-9CEB-4325-BDBC-83A0AC85B8A7) and under the common name Sakhalin toad. We suggest the Chinese common name Dōng Běi Chán Chú for this species. The species was described under the name *B. vulgaris sachalinensis* Nikolsky, 1905 (Type locality: Sakhalin Island, Russia, syntypes ZISP 1934–1936 and MNKNU 26290; *Supplementary file 1A*, *Figure 1—figure supplement 1*). We dated the split within *B. gargarizans* from the Middle Pleistocene (*Figure 5*), and the two species segregated along a diversity of ecological requirements (*Figure 9*). We then propose the assignment of clades in the Korean Peninsula and the Amur River Basin to different subspecies under *B. sachalinensis* to reflect evident geographic and genetic distinction. This framework aligns with the lineage-splitting pattern suggested by Model B in the nuDNA species delimitation analysis which provides a model support value only marginally lower than that of Model A (*Figure 9*), while our mtDNA phylogeny provides robust support for subspecies-level differentiation of these groups. The population on the Korean Peninsula does not have precedence in the taxonomy and it is not the type locality to any *Bufo* that is not synonymous with *B. stejnegeri* (*Supplementary file 1A*, *Figure 1—figure supplement 1*). Hence, we refer to this evolutionarily significant unit as *B. s.* cf. *sachalinensis*.

## Discussion

Our reconstruction of the Holarctic bufonids biogeography using species tree with fossilized birth-death calibration and phylogenetic hypotheses testing verified that the earliest split between Western and Eastern Palearctic *Bufo* occurred during the Middle Miocene (c. 14.46–10.00 Mya; *Table 1*),

temporally matching with the maximum dust outflows in the deserts of Central Asia (*Guo et al., 2002*). Subsequently, we retraced the single-origin Asian *Bufo* in eastern Asia before the split between the continental and Japanese lineages, corroborated by the two Miocene paleogeological events: the rapid uplifts of regions adjacent to the QTP (Hengduan mountains; *Xing and Ree, 2017*) and the drift of the Japanese Archipelago (*Barnes, 2003*).

Our revisit of the three phylogeography hypotheses did not favor any specific hypothesis explaining the present geographic distribution of eastern Palearctic *Bufo,* in disagreement with the long colonization hypothesis (*Zhan and Fu, 2011*). Instead, our results provided support to the combination of the three elements: vicariance of the western mainland clade, followed by long dispersal, and possible refugia in northeastern Asia during the last ice age. We detected a loss of genetic structure within *B. gargarizans* clades, possibly due to introgression resulting from the trade of the species over a millennium (*Lee et al., 2021*). The recent segregation around the Yellow Sea, as shown by the species distribution models and migration pattern, also provides support to the delineation of a septentrional Asian *Bufo* clade associated with range shift toward northern latitudes. We therefore resurrect the previously described *B. sachalinensis* and elevate it to the species level. Our findings resolved the taxonomic boundaries in the *B. gargarizans* complex, and redefine the taxonomic and conservation units: *B. g. gargarizans, B. s. sachalinensis,* and *B. sachalinensis* cf. *sachalinensis*, with the latter waiting for a subspecies description.

## Radiation of Holarctic bufonids

Fossil calibration estimates refined the known divergence time for Neotropical and Eurasian Bufonid lineages and rejected the hypothesis of shared origin (*Sanchiz, 1997*). Although we could not confirm the African biogeographic origin of the Western Palearctic bufonids (*Figure 2*), our time estimates show that the emergence of bufonids into Europe was subsequent to the landmass connection of Eurasia and Africa (*Frazão et al., 2015*). In addition, our ancestral range analysis rejected the hypothesis that bufonids dispersed out of East Asia and into Europe (*Sanchiz, 1997*; *Roček and Rage, 2003*). Instead, we confirm the dispersion of bufonids from Europe and into East Asia, in agreement with the Central Asian desertification (*Garcia-Porta et al., 2012*) as the factor of segregation between the Western and Eastern Palearctic bufonids. The Miocene radiations within the Western Palearctic *Bufo* clade (c. 14.49–11.03 Mya; *Table 1*, *Figure 2*) are consistent with estimates of the highest dusting outflow in the deserts of Central Asia (c. 15.00–13.00 Mya; *Guo et al., 2002*). The desertification may have triggered an early colonization of Asian *Bufo* in central Eastern Asia (c. 9.99 Mya; *Figure 2*), a process that may have taken place prior to the second period of maximum dusting emission in Central Asia (c. 8.00–7.00 Mya; *Guo et al., 2002*).

## Mitonuclear discordance

The phylogenetic trees of East Asian *Bufo* reveal a conspicuous discordance between the topologies of the mtDNA and nuDNA, especially in the placement of regional populations of the *B. gargarizans* complex and the inconsistency in positioning *B. bankorensis* and *B. andrewsi* (*Figure 4*). For instance, the mtDNA phylogeny placed the *B. gargarizans* complex within a nested monophyletic clade, with *B. gargarizans* in the East Asia, the Korean Peninsula, and the Amur River Basin as sister groups to each other (mtDNA tree: Clades 5 and 6; *Figure 4*). Conversely, the nuclear tree did not recover a clear geographic structure for the *B. gargarizans* complex (*Figure 4*). This discordance might reflect the dissimilarity in the evolutionary rates of nuclear and mitochondrial markers as *RAG-1* in amphibians is known to have a slower rate of evolution in comparison to mtDNA (*San Mauro et al., 2004*).

Introgression and incomplete lineage sorting could have contributed to the discordance between mitochondrial and nuclear trees. Here, the D-statistical analysis provides support to the occurrence of introgression resulting in mitonuclear disequilibrium among populations of *B. gargarizans* in the Central East Asian mainland (Patterson's D Statistic =1.0). Introgression is a common occurrence in amphibians, and may be naturally and anthropogenically occurring along contact zones (*Bell and Irian, 2019*). Ancestral polymorphism may have contributed to the incomplete lineage sorting, a point already discussed between *B. gargarizans* and *B. bankorensis* (*Yu et al., 2014*). We recommend testing the possibility of incomplete lineage sorting in contact zones between the parapatric members of the *B. gargarizans* complex: *B. tibetanus*, *B. andrewsi*, and *B. gargarizans* (*Figure 7*).

## Geologically driven divergence in Asia

Past geological events such as the central Asian desertification, the orogeny of the Tibetan plateau, and the Eastern Pacific drift had a more important impact than geographical distribution and selection in shaping the genetic structure of the Eastern Palearctic *Bufo* (*Supplementary file 1C*, *Figure 3*). Selection is a crucial aspect in evolution, however, our model indicates that selection based on life-history strategies had a peripheral role in influencing the evolutionary pathways of Asian bufonids. While the relationship between Asian *Bufo* and the species previously assigned to '*Torrentophryne*' remains uncertain (low support of p=0.03; *Figure 3*), the most accurate species tree topology rejects the monophyly of the lotic breeders '*Torrentophryne*' (Model C vs. Model E; *Supplementary file 1C*, *Figure 3*). The paraphyletic nature of '*Torrentophryne*' is also corroborated by previous taxonomy studies (*Liu et al., 2000*; *Pyron and Wiens, 2011*). Nevertheless, our topology models validate the convergence among semi-aquatic members of Asian *Bufo* (*B. andrewsi, B. stejnegeri,* and *B. torrenticola*) and the homoplasy between *Torrentophryne* and *Bufo* (*Supplementary file 1C*, *Figure 3*).

## Miocene to Pleistocene diversifications in Asia

Subsequent to the early radiation in association with the Central Asian dust events at the end of the Middle Miocene, we confirmed a single origin of Asian *Bufo* from Eurasia through Central Asia during the Late Miocene (*Supplementary file 1G*, *Figure 5*). The Asian *Bufo* shared an ancestor before the divergence between the East Asian mainland and Japanese Archipelago lineages took place (*Figure 5*) as a result of the two most important vicariance events that occurred contemporarily during the Miocene (*Figure 5A1-A2*). Our dating estimates, previous hypotheses (*Igawa et al., 2006*; *Macey et al., 1998*) and paleogeological events are consistent for two main points: (i) an early radiation of the Asian continental *Bufo* triggered by the QTP vicariance (Miocene; c. 8.36 Mya; *Table 3*, *Supplementary file 1G*, *Figure 5A2*) and (ii) the isolation of *Bufo* clades on the Japanese archipelago (Late Miocene; c. 7.71 Mya; *Table 3*, *Supplementary file 1G*, *Figure 5A1*). The early radiation of the Asian continental *Bufo* occurred in high elevation-restricted clades: *B. tibetanus* and *B. andrewsi* (*Figure 5*) and coincides with the pre-completion of the orogenesis of the Hengduan Mountain (Late Miocene; c 8.0–10.0 Mya; *Xing and Ree, 2017*). The estimated time of isolation of the Japanese *Bufo* from the Asian continental clades (c. 7.71 Mya; *Table 3*, *Supplementary file 1G*, *Figure 5*) is pertinent, as the event may have occurred before the complete separation of the Japanese Archipelago from the Eurasian landmass (c. 5.00 Mya; *Barnes, 2003*).

The major radiations within the primary clades of East Asian mainland *Bufo* took place predominantly between the Pliocene and Pleistocene (*Figure 5A3*), with dispersals principally driving the diversifications of the *B. gargarizans* complex clades in the lowlands (*Figure 5*). In comparison, divergence in the insular and septentrional East Asia clades was mostly driven by vicariance and dispersals (clades of Japanese *Bufo* and *B. sachalinensis*; *Figure 5*). In the case of *B. stejnegeri*, although the estimated timing is more recent (Pliocene; c. 3.31 Mya; *Table 3*, *Supplementary file 1G*, *Figure 5*) relative to the estimates in the literature (Pliocene; c. 4.30 Mya; *Fong et al., 2020*), our results are in agreement on the independent rise of this clade on the Korean Peninsula prior to complete formation of the Baekdu Mountains shields (2.80–1.50 Mya; *Kim et al., 2017*).

## The Yellow Sea as a biogeographic boundary

Previous studies estimated the isolation of *Bufo* populations on the Korean Peninsula to date from the Pleistocene (*Borzée et al., 2017*; *Lee et al., 2021*), and our results confirm this pattern. However, additional samples from further east demonstrated the presence of a deeper segregation between the clades around the Yellow Sea, with *B. gargarizans* to the southwest and *B. sachalinensis* in septentrional Eastern Asia (*Figure 6*). This pattern is visible through both mtDNA and nuDNA. A late colonization of the septentrional East Asia regions has also been supported by an increase in population size in relation to the LGM, a pattern not visible for the *B. gargarizans* clades (*Figure 6*). The isolation of *B. sachalinensis* clade on the Korean Peninsula is in synchrony with the final stage of the Sanduo Event that triggered the widening of the Yellow Sea basin (Neogene-Quaternary; *Lei et al., 2018*). Most probably, the drift between clades was subsequently induced by variations in water depth, temperature, and the tidal circulation of the Yellow Sea (*Li et al., 2016*). This pattern is also visible in other amphibian clades that became isolated during the quaternary (*Borzée et al., 2020*).

## Delineation of climatic niche selection

In congruence with the best species delimitation model (*Figure 9*), the niche overlap statistics demonstrated the segregation of the two clades tested: *B. sachalinensis* in septentrional East Asia and *B. gargarizans* in eastern central Asia (or southwestern margin of the Yellow Sea; *Figure 7*). The response curves of the Maxent Jackknife test indicated that the two clades have different temperature-related environmental requirements (*Figure 7—figure supplement 1*) with the monthly and quarterly minimum temperatures being the key factors segregating the niche of the septentrional Eastern *B. sachalinensis* from *B. gargarizans* (see BIO 6 and 11; *Figure 7—figure supplement 1*). The spatial heterogeneity demonstrated by both clades may have led the species to adapt locally, resulting in specific niches, as exemplified in closely related bufonids: *B. bufo* from the Western Palearctic (*Luquet et al., 2015*). Local adaptation may also derive from the apomorphic trait related to behavioral and phenotypic plasticity, as seen in the case of bufogenin production in *B. gargarizans* along a latitudinal gradient with different climates (*Cao et al., 2019*).

## Ineffective gene flow on the septentrional East Asia

Low gene flow rates among the septentrional East Asian clades (*Figure 8*) indicate a strong effect of genetic drift, especially in the isolated clade of *B. s.* cf. *sachalinensis* in the Korean Peninsula. The latest glaciations may not have induced range shifts as the clade was already isolated on the peninsula following the rising level of the Yellow Sea. In addition, low migration rates support the natural selection and gradual adaptation of the *B. s. sachalinensis* clade to the cold climate of the northern latitudes. Long-distance dispersal had a weak impact on *B. s. sachalinensis* population in the Amur River Basin (*Figure 8*), probably due to the reduced capacity for dispersal to compensate for the energy investment on low-temperature tolerance (*Kolbe et al., 2010*).

This pattern of unidirectional gene flow may however need to be seen from a different perspective. The lack of gene flow here may also highlight that the *Bufo* trade may not have been as widespread on the Amur River Basin and Korean Peninsula as it was in China, a pattern correlated with the size of human settlements. The presence of two individuals within the range of *B. gargarizans* showing 95% and 87% admixtures with the *B. sachalinensis* clade (*Supplementary file 1F*, *Figure 4D*) is likely the result of human-led translocations. *Bufo* toads have been used for traditional Chinese medicine for more than a millennium (*Zhan et al., 2020*; *Lee et al., 2021*) and gene flow toward the regions relying on the genus for medicine may actually reveal the directionality of the trade, and escape of individuals at markets, rather than natural dispersal.

## Taxonomic recommendations

The distinction among *B. tibetanus, B. minshanicus,* and *B. andrewsi* cannot be assessed in detail here due to the scarcity of samples. However, *B. tibetanus* and *B. andrewsi* are genetically meaningful as species, each clade being monophyletic and part of a sister clade to *B. gargarizans* (*Fu et al., 2005*). We however note the possible presence of *B. andrewsi* in Zhongdian, the type locality of *B. tibetanus*. Here, we resolve the long-standing question about the relationship between the *B. gargarizans* complex, *B. andrewsi,* and *B. bankorensis* (*Chen et al., 2013*; *Fu et al., 2005*). Despite synonimization between *B. andrewsi* and *B. gargarizans* through mtDNA phylogeny and allozymes data (*Fu et al., 2005*), the coalescent perspective on delimiting the species in the present study supports the taxonomy validity of *B. andrewsi* as a distinct species, a point strongly supported by morphometric data (*Liao et al., 2016*). Here, we demonstrated the impact of the QTP orogenesis on the divergence of *B. andrewsi* from *B. tibetanus* during the Late Miocene (*Table 3*, *Figure 5*). Both paleogeography and species delimitation patterns justify the species epithet, and the clade should be referred to as *B. andrewsi Schmidt, 1925* in later studies. We recommend further research on the relationship with the clade described as *B. tibetanus* in light of the Late Miocene divergence between these two clades (*Figure 5*).

The improved sampling resolution throughout East Asia, combined with the use of advanced taxonomic tools, helped resolve the long-standing cryptic boundaries between *B. gargarizans* and *B. sachalinensis* (*Matsui, 1986*). The splitting of *B. sachalinensis* from *B. gargarizans* results in the newly elevated species containing a single evolutionary lineage (*Figure 9*). This suggests that the species adapted to the local environment after dispersion from glaciation refugia and colonization toward northern latitudes during the Late Pleistocene (*Figures 5 and 9*). A similar pattern of northward

gradient colonization was observed in other amphibians in the area such as *Kaloula borealis* (**Othman et al., 2021**). The presence of two distinct clades, *B. sachalinensis* cf. *sachalinensis* and *B. s. sachalinensis,* highlights a strong allopatric structure in the septentrional East Asia clades rather than a case of recent divergence (**Figure 4**), highlighting the significance of segregated conservation units for these two lineages.

We corroborate the validity of *B. bankorensis* as an independent species through the species delimitation analysis (**Figure 9**), contemporaneous with the divergence time of other clades within the *B. gargarizans* complex, and an active period of glaciations (Mid to Late-Pleistocene; **Figure 5**). This clarifies the presence of a common ancestor, followed by a single evolutionary lineage for each of the clades within the *B. gargarizans* complex, as well as *B. bankorensis* on Taiwan Island. As supporting evidence, the niche segregation between *B. bankorensis* and the other *B. gargarizans* clades is elevation related. *Bufo bankorensis* is distributed in mountainous areas (**Lin and Lue, 2004**), *B. gargarizans* in the lowlands, and *B. minshanicus* at higher elevation on the QTP, sympatric with *B. andrewsi* (**Fu et al., 2005**).

Despite the current support on the splitting of *B. j. formosus* and *B. j. japonicus* in relation to the Pliocene-Pleistocene divergence (**Figures 5 and 9**), we do not recommend further taxonomic changes as the taxonomy of the clades is already resolved (**Dubois and Bour, 2010**; **Matsui, 1984**), and the type locality of the later subspecies is not restricted to any specific area in Japan. We do however provide evidence about the independent evolution of the two subspecies of *B. japonicus* through historical events and species delimitation, which could be used as the phylogenetic support needed for the subspecies to be elevated to the species level.

It is worth pointing out the widespread risk for most *Bufo* species studied in this work. Anthropogenic introductions have already resulted in threats to insular populations such as *B. j. formosus*, which suffered from introgression with *B. j. japonicus* (**Hase et al., 2013**). Emphasis should be placed on limiting the trade to regions within the geographic occurrence of the clade traded to avoid such risks, and especially for isolated clades with high genetic homogeneity and small genetic variability, such as *B. s.* cf. *sachalinensis* ($\pi$=4.76±2.34; **Supplementary file 1D**). A risk reflected through the declining effective population size in the Korean population compared to the East Chinese one (**Figure 6**). In addition, trading *Bufo* species can result in artificial introduction, as seen with the introduction of *B. j. formosus* on Hokkaido (**Suzuki et al., 2020**) and the spread of pathogens (**Borzée et al., 2021**).

In summary, ever since the Miocene and throughout the ice ages, past paleogeological events have been the most powerful factors promoting the genetic structures of *Bufo* in the Eastern Palearctic. A segregation between the West and East Palearctic *Bufo* clades in relation to the Central Asian dusting resulted in the vicariances of Asian *Bufo*, mainly under the influences of the QTP orogenesis and the Japanese continental drift. Unlike the simple allopatric speciation, co-distribution and artificial secondary contact have resulted in a blurry genetic structure of the clades inhabiting the central East Asian lowlands: *B. gargarizans gargarizans* and *B. g. popei* both showcase the likely loss of genetic diversity during the Anthropocene. Glaciations impacted the range expansions of *Bufo* to septentrional East Asia, and led to microclimate adaptation along the latitudinal gradient of populations of the *B. gargarizans* complex. As a long-sought resolution to long-standing systematics problems, we elevated *B. sachalinensis* from septentrional East Asia to the species level. This recommendation is supported by the population expansion resulting from the LGM, the gene flow toward northern latitudes, and a significant distinction in niche suitability when compared with *B. gargarizans*. Concurrently, the revision identified two significant conservation units linked to the subspecies *B. s. sachalinensis* in the Amur River Basin and *B. s.* cf. *sachalinensis* on the Korean Peninsula. Future research on behavior and call properties could further elaborate on the differentiation highlighted here with genetic tools, and genomic studies may help confirm the genetic patterns demonstrated herein. In addition, this study insists on a strong call for the genetic preservation of Asian *Bufo* clades to prevent future loss of genetic diversity.

## Materials and methods
### Taxa sampling and data set design
We sampled 274 *Bufo* individuals across East Asia. We supplemented our data set with Genbank sequences to cover the totality of the described range of the species, from the northern distribution

in Sakhalin Island to the southern distribution limit in Vietnam. The specimens collected represent four recognized species and two subspecies: *B. andrewsi, B. gargarizans, B. stejnegeri, B. bankorensis, B. japonicus japonicus,* and *B. j. formosus* (*Supplementary file 1J*). In addition, we used sequences from Genbank (N=158; *Supplementary file 1J*), notably adding samples listed as *B. tibetanus* and *B. minshanicus* from western China and a sample of *B. gargarizans* from Vietnam (*Supplementary file 1J*). We selected five loci from the *CR*, mitochondrial protein-coding *ND2,* and nuclear protein-coding markers *POMC, RAG-1,* and *Rho*. To amplify these targeted gene fragments, we adapted universal primers used in closely related studies (*Supplementary file 1K*). In addition, we designed primer pairs for several gene fragments (*Supplementary file 1K*), modified from the most homologous sequences in Genbank using Primer 3web v.4.1.0 (*Untergasser et al., 2012*). Details on the molecular works such as the isolation of gene fragments and the polymerase chain reaction (PCR) amplifications are explained (Appendix 1) with the final primers used and the PCR conditions (*Supplementary file 1K*).

From these samples, we designated three data sets varying in number of taxa and geographical scale (*Figure 1*), with considerable overlap of samples. We described the methodology and the data sets as follows: (1) estimates of divergence dates in the Holarctic bufonids species tree using a fossil-ized birth-death model (*Heath et al., 2014*) and estimating divergences to a time frame coherent with the minimum fossils age. For the species tree reconstruction, we included additional sequence data from two loci, a mitochondrial ribosomal large subunit 16S rDNA (*16S*) and a nuclear protein-coding C-X-C chemokine receptor type 4 (*CXCR4*) available in Genbank. Here, the final data set comprised seven unlinked molecular loci obtained from gene fragments of *CR-16S-ND2-CXCR4-POMC-RAG-1-Rho* (2666 bp) and included 313 sequences belonging to 39 taxa distributed across South and Central America, Nearctic, and Eurasia (*Supplementary file 1J*). (2) Evaluation of the best species tree topology for the Palearctic *Bufo* species under a hypothesis testing approach derived from geograph-ical, geological, ecological, and life-history variables. In this second data set, we assembled 2959 bp of unlinked multi-locus gene fragments (*CR-16S-ND2-CXCR4-POMC-RAG1-Rho*) from 267 individuals of 24 recognized taxa (*Supplementary file 1J*). (3) Estimation of the divergence dates and reconstruc-tion of the ancestral ranges of the Eastern Palearctic *Bufo* genus from mtDNA and nuDNA sequences data. Here, we refer to the unlinked mtDNA (*CR-ND2*: 894 bp; N=221) as data set 3a and the unlinked nuDNA data (*POMC-RAG-1-Rho*: 1030 bp; N=44) as data set 3b. We based the time tree calibrations on primary and secondary calibrations, using a combination of fossil-based estimates, paleogeolog-ical events, and literature data under various combinations of molecular clocks and tree priors (see detailed calibration points in Appendix 1).

## Reconstruction of Holarctic bufonids biogeography

We calibrated the species tree under the fossilized birth-death process for 39 recognized species of Holarctic bufonids. To do so, we included species groups with documented fossils or close fossils rela-tives. To estimate the divergence date, we employed a relaxed molecular clock model with the fossil-ized birth-death and Yule as tree priors. We then determined the best site model for each unlinked marker (*Supplementary file 1K*) using jModelTest v.2.1.10 (*Darriba et al., 2015*). We relied on six calibration points based on the age ranges of the fossil records, and assumed the age as minimum constraint (*Donoghue and Yang, 2016*).

We retrieved the documented fossil record for all Holarctic bufonids from the database of verte-brates (FosFARbase, accessed June 1, 2020; URL: https://www.wahre-staerke.com/) to determine our calibration points. We then verified the accuracy of each fossil's age range by cross-referencing the literature (*Supplementary file 1B*). To ensure the reliability of the calibration, we excluded the commonly used Nearctic fossil of '*Bufo*' *praevius* (20 Mya; *Supplementary file 1B*) due to known ambiguities (*Pauly et al., 2004*) and the only conditional attribution of this species to the genus. Instead, we relied on the approximation of the age of their oldest fossil relatives (*Rhinella* of Southern America; *Supplementary file 1B*). The detailed descriptions for each calibration point are further elaborated in the Appendix 1.

We performed four independent analyses using the Markov Chain Monte-Carlo (MCMC) proce-dure for 200 million iterations using StarBEAST with the Relaxed Clock (*BEASTRLC) package installed through BEAST v.2.6.3 (*Bouckaert et al., 2019*). We ensured the adequacy of the MCMC samplings by assessing the effective sample size (ESS) values of each parameter (ESS>200) using Tracer v.1.7 (*Rambaut et al., 2018*). We assembled all the generated trees in LogCombiner v.2.6.1 (*Bouckaert*

*et al., 2019*) and summarized the trees with a Maximum Clade Credibility (MCC) tree after discarding 25% of the trees with a posterior probability limit of 0.5 using Tree Annotator v2.6.3 (*Bouckaert et al., 2019*).

## Inferring patterns of diversification of Eastern Palearctic *Bufo*

### Species tree topology estimation for Palearctic *Bufo*

We estimated the most probable topology of the species tree for the Palearctic *Bufo*. We emphasized the inclusion of all optimum factors that control the topology of the tree, we also included species from the former genus *Torrentophryne*, which is currently synonymized with East Asian *Bufo* in our data sets.

We tested five alternative scenarios for the species tree topology, derived from relevant geological, ecological, and life-history data associated with the phylogeny of East Asian bufonids. The factors shaping the tree topology model for each species and detailed descriptions are as such: (1) geographic range and isolation by distance, (2) life history (i.e., terrestrial or semi-aquatic) and adaptive morphological traits (i.e., visible tympanums vs. lacking of external tympanum, presence or absence of adhesive abdomen in tadpoles), (3) a single origin for the eastern Palearctic *Bufo* lineages in relation to past geological events (i.e., QTP Miocene orogenesis and continental plates shifts; *Igawa et al., 2006*; *Macey et al., 1998*), (4) alternative topology 1: two independent origins to the Eastern Palearctic *Bufo*: the East Asian mainland *Bufo* vicariance followed by '*Torrentophryne*' radiations around the QTP regions, and the Japanese *Bufo* divergence directly from the Western Palearctic ancestor through colonization before the Miocene Pacific drift, and (5) alternative topology 2: a single origin of the Eastern Palearctic *Bufo* from the QTP vicariance resulting into the following monophyletic clades: the East Asian mainland *Bufo*, the Japanese *Bufo*, and high elevation '*Torrentophryne*' group in Tibet.

Here, we set *Bufotes* as our external calibration point. Four internal calibration dates for ingroup taxa (*Bufo* genus) were adapted from the time frame we determined during the previous step for the Holarctic bufonids data set using the fossilized birth-death method. We employed a relaxed molecular clock and Yule prior for each designated species tree data set. We ran two independent analyses with MCMC samplings for 30 million iterations using the *BEASTRLC package (*Heled and Drummond, 2010*) implemented in BEAST v.2.6.3 (*Bouckaert et al., 2019*). The convergence of the MCMC runs of each analysis were assessed by the ESS values obtained for each parameter (ESS>200) using Tracer v.1.7 (*Rambaut et al., 2018*). Finally, we evaluated all the supports for species tree topology models with the nested sampling analyses using the nested sampling (NS) package (*Russel et al., 2019*) installed through BEAST v.2.6.3 (*Bouckaert et al., 2019*). We set the sampling parameters as follows: particle count = 20, the length of MCMC subchains = 20,000, and epsilon = $10^{-6}$.

### Phylogenetics and divergence dating of Asian *Bufo*

To reconstruct the phylogenetic relationship, we searched for the best-fit evolutionary model and partitioning scheme for each gene data set (*Supplementary file 1M*) using PartitionFinder v.2.1.1 (*Lanfear et al., 2017*). We evaluated six partitions for the concatenated mtDNA data set, considering three partitions represented a *CR* fragment and two partial introns of *ND2*, and three codon reading frames for exon of *ND2* (*Supplementary file 1M*). We adjusted the setting to suit 'Mr. Bayes' mode, using the greedy algorithm. We selected optimal partitioning schemes, and the best-fit substitution models using the Bayesian Information Criterion (BIC). We implemented the best-fit model for each sequence data based on the suggested partitions.

In order to evaluate the recombination pattern in the nuDNA sequences, we conducted a phi test analysis on the concatenated sequences of 1030 bp of *POMC-RAG-1-Rho* (N=44) using Splitstree v.4.14.6 (*Huson, 1998*). The phi test did not statistically support the presence of significant recombination (p=0.82) in the nuDNA data set and the data was therefore reliable for further phylogenetic analyses. We obtained the best-fit evolutionary model for the 11 partitions, evaluated on each partial intron of *POMC* and *Rho*, and three codons position for each exon of *POMC*, *RAG1*, and *Rho* (*Supplementary file 1M*).

For both mtDNA and nuDNA data sets, we analyzed the haplotypes with median-joining method (*Bandelt et al., 1999*) and inferred the network using POPART v.1.7 (*Leigh and Bryant, 2015*). We reconstructed concatenated genes trees with Bayesian Inference (BI) method for 100 million of MCMC chains with 20% burn-in using Mr. Bayes v.3.2.6 (*Huelsenbeck and Ronquist, 2001*), until the run

reached convergence with a standard deviation (SD) values lower than 0.05. We repeated the reconstruction of the trees following the maximum likelihood (ML) approach using IQ-TREE implemented in the web version of W-IQ-TREE (http://iqtree.cibiv.univie.ac.at) (*Trifinopoulos et al., 2016*). To infer the population-based structure, we ran population genetic statistical analyses such as genetic differentiation, neutrality tests, and hierarchical analysis of molecular variance (AMOVA) on the mtDNA data (Appendix 1). We inferred the potential genetic cluster (K) on the nuclear data and further examined the introgression patterns following population admixture analysis (Appendix 1).

Then, we estimated the divergence dates of the East Asian *Bufo* clades on the unlinked data of mtDNA (data set 3a) and nuDNA (data set 3b) independently. We first examined the most appropriate molecular clock to be used for the data sets (Appendix 1, *Supplementary file 1N*). As a result, we employed a relaxed molecular clock model for the divergence dates estimates with the mtDNA data set, and a strict clock model for the nuDNA data set (*Supplementary file 1N*). We reconstructed the time tree based on the fixed topology obtained from the phylogenetic relationship through ML and BI trees. For the mtDNA data, the topology of the time tree was also consistent with the previous study (*Borzée et al., 2017*). We set an external calibration point based on the fossil of *Bufotes* as outgroup, and seven internal calibration points for the focal clades (ingroup), relying on combined evidences from fossil records, paleogeological events, and secondary calibration from the literature. We relied on a log-normal distribution with real space mean for the calibrations of fossil records, and we used a normal distribution for the calibration of the paleogeological events and the secondary calibrations from the literature. The enforced internal calibrations are described in more detail (Appendix 1).

To reduce bias in prior choice, we tested the mtDNA time tree using the same seven internal calibration points and different combinations of relaxed clocks and tree priors such as Yule and birth-death. We performed all divergence dating analyses in BEAST v.2.5.2 (*Bouckaert et al., 2019*). To maintain consistency, each MCMC analysis was run for 30 million generations, with two independent runs. We diagnosed the stationary states of all parameters using Tracer v.1.7 (*Rambaut et al., 2018*). We then used tree annotator (*Rambaut et al., 2018*) to sort a MCC tree to summarise all trees generated with a median height, after 25% burn-in and 0.5 as minimum posterior probability. We further inferred the historical biogeography of the genus to test possible ancestral areas for each clade using the Bayesian dispersal-vicariance (BBM-DIVA) in RASP v.3.1 (*Yu et al., 2015*).

## Past population dynamics

We tested the hypothesis of an ice age refugia (*Borzée et al., 2017*) against the earlier hypotheses of vicariance during the QTP followed by dispersal (*Macey et al., 1998*) and the hypothesis of dispersion through long dispersal (*Fu et al., 2005*). We assessed support for these hypotheses by examining past population dynamics for the two distinct mtDNA *B. gargarizans* clades across a major biogeography barrier, the Yellow Sea. By doing so, we also gained a phylogeographic insight into the impact of the Yellow Sea on the demographic history of *B. gargarizans* during the ice age.

We divided our data into two sets following the clades identified by the mtDNA phylogenetic tree. These clades, hereafter referred to as the southwestern (N=75) and the septentrional East Asia (N=123) clades, were also geographically segregated by the Yellow Sea margins. To estimate the past population dynamics since the LGM period, we constructed for each clade a Bayesian Skyline Plot (*Drummond et al., 2005*) inferred from the 819 bp of concatenated *CR-ND2* using BEAST v.2.5.2 (*Bouckaert et al., 2019*). We calibrated the data sets with a combination of strict molecular clock and the Coalescent Bayesian Skyline as prior, under a lognormal distribution. To calibrate the southwestern clade, we set the mean in the real space to 2.5, SD to 0.5, and offset to 0.0, creating an interval of HPD 5%–95% that ranged between 0.96 Mya and 5.02 Mya. For the calibration of the septentrional East Asian clade, we set the mean in the real space to 0.8, SD to 0.5, and offset to 0.0, creating an interval of HPD 5%: 0.30 Mya to 95%: 1.84 Mya. We ran two independent analyses on each data set with 5 million MCMC iterations and a burn-in of 1000 samples. We looked for the stationary state of the runs and evaluated the ESS values obtained for all the parameters with Tracer v.1.7 (*Rambaut et al., 2018*), before plotting the Bayesian Skyline Plot with the same program.

## Ecological niche modeling

Given the support for two clades within the *B. gargarizans* complex, here named *B. gargarizans* and *B. sachalinensis*, we determined the boundaries of each clade through ecological niche modeling.

First, we determined the niche overlap between the two clades, and computed habitat suitability models for each clade using maximum entropy (MaxEnt) modeling (*Phillips et al., 2017*). To build our model, we combined the geographic information linked to the genetic data and occurrence data downloaded from GBIF (GBIF.org, accessed August 27, 2020; DOI: https://doi.org/10.15468/dl.ugtsma). We used only data points from the Asian mainland, and manually checked each datapoint. We then assigned each point to either *B. gargarizans* or *B. sachalinensis* based on the geographic distance to the nearest locality with genetic data, resulting in 404 occurrences for *B. gargarizans* (375 GBIF, 29 genetics) and 1076 occurrences for *B. sachalinensis* (1053 GBIF, 23 genetics). Three genetic data points belonging to the septentrional East Asian clade were located within range of the western mainland clade. These geographically discordant datapoints are likely the result of human introduction, only the data point assigned to the nominal clade was used for modeling.

To build the models, we used 19 bioclimatic variables (*Supplementary file 1O*) and three terrain variables extracted from a digital elevation model (DEM; United States Geological Survey) including elevation, slope, and mean slope within a grid cell of 12.27 km by 12.27 km. The spatial resolution for all environmental layers was 0.0417 decimal degrees, or approximately 4.1 km. Duplicate presence points (points in the same grid cell in environmental layers) were removed as an option in the MaxEnt model platform to reduce spatial bias, resulting in a final sample size of 275 independent datapoints for *B. gargarizans* and 468 independent datapoints for *B. sachalinensis*. We ran MaxEnt for 10 bootstrap replicates with a random test percentage of 20%, and we used the averages of all replicates as final models. Here, we decided to include all 22 layers, despite the risk of correlation, as we needed to use the same variables for the two clades for downstream analyses and to avoid the exclusion of relevant variables, based on the previous method (*Borzée et al., 2020*). Models were evaluated using area under the curve (AUC) and true scale statistic (TSS) (*Allouche et al., 2006*). The niche overlap of the models was evaluated using the niche overlap function in the 'dismo' package (*Hijmans et al., 2015*) in R version 3.5.1 and RStudio version 1.1.442. Both 'I' and 'D' statistics were calculated.

To determine whether niche overlap of the two *Bufo* clades was significantly different than expected if the two clades had the same environmental requirement, we created a null distribution using MaxEnt modeling. To simulate randomized occurrences, we pooled occurrences from both clades and then randomly assigned individuals to *B. gargarizans* (404 points) or *B. sachalinensis* (1076 points). We then ran MaxEnt using the same parameters as described above for 20 runs and calculated 'I' and 'D' overlap statistics between corresponding pairs of suitability outputs. This created a null distribution that we then tested against the niche overlap values calculated from the initial run separating the two clades using a one-sample Student's t-test. Finally, we reconstructed maps in ArcMap 10.6 (ESRI, Redlands, CA).

## Estimation of migration patterns

Finally, we estimated the impact of evolutionary forces such as gene flow on the population structure of the septentrional East Asian clades. In order to measure the magnitude and direction of gene flow, we carried out a migration test on a total of 60 *Bufo* individuals representing three subpopulations in the northern latitudes herein defined as: northeastern mainland (N=12), Korean Peninsula (N=24), and Amur River Basin (N=24). Following the assumption that the mutation rate varies among loci and migration is asymmetrical between population, we computed the migration pattern from the following unlinked loci *CR* (894 bp), *ND2* (536 bp), *POMC* (496 bp), and *RAG-1* (308 bp). We set the mutation rate to vary among the loci and standardized the migration rate as 4 N m =1.0.

In addition, we estimated the migration pattern of *B. stejnegeri* on the Korean Peninsula, between the northern and southern populations. This analysis was based on 12 individuals sampled north (N=9) and south (N=3) of the Republic of Korea and aimed at testing whether migration is significantly higher in one direction than the other. We conducted this analysis using a full migration matrix model in MIGRATE v.4.4.3 (*Beerli et al., 2019*). We computed two independent analyses for each locus following the MCMC method for 10 million iterations, with four parallel chains and a burn-in of 1000. We evaluated the ESS value of all parameters sampled for each analysis with Tracers v.1.7 (*Rambaut et al., 2018*).

## Species delimitation modeling

Due to the uncertainty in the taxonomic placement of many East Asian Bufonids, we examined the species boundaries of the Asian *Bufo*, with emphasis on the five putative species within the *B. gargarizans* species complex. We tested different species delimitation scenarios such as sympatric ranges and intergradation zone between subspecies. To account for human-led population displacements, we excluded *B. gargarizans popei* from the species delimitation model itself, and lumped it with *B. gargarizans gargarizans* under *B. gargarizans*. We relied on synonymized subspecies clades to define geographic areas (*Pyron and Wiens, 2011*); hence, the designated models included all clades of the *B. gargarizans* complex in the mainland with unresolved taxonomy: *B. andrewsi*, *B. bankorensis*, *B. gargarizans* from the central East Asian mainland of its range, and septentrional East Asian *B. gargarizans*. We also included the two Japanese *Bufo* subspecies, *B. j. formosus* and *B. j. japonicus*, as indicated by the two distinct clades in our phylogeny, and the Korean water toad, *B. stejnegeri*. We used previously named clades that are now synonymized to test all models possible and reintroduced synonymized names for the purpose of referring to these clades in our models.

We justified the use of these names following the earliest taxonomically valid binomial name (for which type specimen and type locality have been recorded; *Supplementary file 1A*, *Figure 1—figure supplement 1*). The clade of septentrional East Asian *Bufo* distributed on the Korean Peninsula is segregated from *B. gargarizans* (*Matsui, 1986*), and we deviated from the name '*asiaticus*' originating from the described species '*B. vulgaris* var. *asiaticus*' Steindachner, 1867 as the type locality for the "*asiaticus*" clade is located within the range of *B. gargarizans* (type locality: Shanghai, China; *Supplementary file 1A*, *Figure 1—figure supplement 1*). We leave the Korean clade unnamed, hereafter referring to it as '*B. sachalinensis* cf. *sachalinensis*.' The clade distributed in the Amur River Basin (referred to as Far Eastern Russia in other works) was referred to as *B. sachalinensis sachalinensis*, in reference to '*B. vulgaris* var. *sachalinensis*' Nikolsky, 1905 (type locality: Sakhalin Island, Russia; *Supplementary file 1A*, *Figure 1—figure supplement 1*). See the Results section for taxonomic resolutions regarding the justification of these names.

We designed eight competing models for species delineation derived from 1030 bp of concatenated nuDNA (*POMC-RAG-1-Rho,* N sequence =16). We set the species delimitation model under splitting or lumping scenarios. We tested the presence of a single evolutionary lineage for disputed clades within the *B. gargarizans* complex in the mainland, such as the validity of *B. bankorensis* and *B. andrewsi*. We also evaluated the species boundary between the cryptic clades of *B. sachalinensis* cf. *sachalinensis* and *B. s. sachalinensis* on the Korean Peninsula and the Amur River Basin, respectively. Detailed descriptions of the eight alternative species delimitation models are provided in *Supplementary file 1I*.

Using a coalescent-based approach, we ran the eight species delimitation data sets under the Bayes Factor Delimitation (BFD) method with SNAPP Package (*Bryant et al., 2012*) implemented in BEAST v.2.5.2 (*Bouckaert et al., 2019*). We calculated the coalescent rates and set the mutation rate to 53.66 (U) and 0.504 (V). We then chose gamma mutation models with log-likelihood correction selected. We set the lambda prior based on the number of taxa in the designated model. We ran each data set separately with two independent analyses before fixing the final parameter in the operator. We ran the analyses with the MCMC procedure to 1 million generations and we sampled every 1000. We evaluated the convergence state of each model by ensuring that the effective samples size (ESS value) obtained was higher than 200 for each parameter using Tracer v.1.7.1 (*Rambaut et al., 2018*).

To compare the support of each species delimitation model, we then ran a stepping stone/path sampling analysis with Path sampler in the Model Selection Package (*Russel et al., 2019*) installed in BEAST v.2.5.2 (*Bouckaert et al., 2019*). For each data set, we used 0.5 million chains length and eight steps in the paths. We computed the path sampling analyses with the MCMC procedure for 1 million generations until obtaining the MLE value. We determined the rank of support for each species delimitation model using the MLE and Bayes factor values. We calculated the Bayes factor with the formula: Bayes factor = 2×(MLE value of model $X_0$)–(MLE value of model $X_1$). The species delimitation model with a positive Bayes factor value was the most favorable model, relative to models with negative Bayes factor value.

## Acknowledgements

The authors thank Sungsik Kong, Jordy Groffen, Yi Yang, Vasiliy I Kazakov, Ye Inn Kim, and Sera Kwon for their commitment and help during fieldwork. The authors also thank Hakyung Kang for the keen assistance in obtaining sampling permits from local authorities. The authors credit Christophe Dufresnes, Antonio Rodríguez Arduengo (*Rhinella marina*), David Leo Berton (*Bufo verrucosissimus*), and Ben Zerante (*Bufo japonicus*) for the photos of toads used in Figure 1. The authors acknowledge Professor Yeong-Choy Kam of Tunghai University for providing logistic helps during fieldwork. The authors also thank Jonathan J Fong and an anonymous reviewer for their comments during the review.

## Additional information

### Funding

| Funder | Grant reference number | Author |
|---|---|---|
| Korean Environmental Industry and Technology Institute | KEITI 2017002270003 | Yikweon Jang |
| Foreign Youth Talent Program | QN2021014013L | Amaël Borzée |
| Russian Foundation of Basic Research | 20-04-00918 | Spartak N Litvinchuk |
| Portuguese Foundation for Science and Technology (FCT) | SFRH/ BPD/109148/2015 | Michael J Jowers |

The funders had no role in study design, data collection and interpretation, or the decision to submit the work for publication.

### Author contributions

Siti N Othman, Conceptualization, Data curation, Formal analysis, Investigation, Methodology, Software, Validation, Visualization, Writing – original draft, Writing – review and editing; Spartak N Litvinchuk, Conceptualization, Data curation, Resources, Validation, Visualization, Writing – review and editing; Irina Maslova, Project administration, Resources, Validation, Writing – review and editing; Hollis Dahn, Data curation, Resources, Validation, Writing – review and editing; Kevin R Messenger, Data curation, Resources, Writing – review and editing; Desiree Andersen, Formal analysis, Validation, Writing – review and editing; Michael J Jowers, Data curation, Resources, Validation, Visualization, Writing – review and editing; Yosuke Kojima, Dmitry V Skorinov, Kiyomi Yasumiba, Ming-Feng Chuang, Yi-Huey Chen, Yoonhyuk Bae, Jennifer Hoti, Resources, Writing – review and editing; Yikweon Jang, Funding acquisition, Project administration, Resources, Writing – review and editing; Amael Borzee, Conceptualization, Investigation, Methodology, Project administration, Resources, Supervision, Validation, Visualization, Writing – review and editing

### Author ORCIDs

Siti N Othman http://orcid.org/0000-0002-3894-7553
Michael J Jowers http://orcid.org/0000-0001-8935-5913
Ming-Feng Chuang http://orcid.org/0000-0002-7328-2577
Yi-Huey Chen http://orcid.org/0000-0003-0987-2385
Yikweon Jang http://orcid.org/0000-0002-7213-7319
Amael Borzee http://orcid.org/0000-0003-1093-677X

### Ethics

Sampling in the Republic of Korea were collected in 2017 under the Ministerial authorisation number 2017-16, and the samples from Jirisan National Park were collected under the Ministerial authorisation number 2019-01. Samples from the People's Republic of China were collected under the authorisation provided by Nanjing Forestry University. IACUC permit is not required for the in-situ experiment in this study, in accord to the rules of Ewha Woman's University Institutional Biosafety Committee.

**Decision letter and Author response**
Decision letter https://doi.org/10.7554/eLife.70494.sa1
Author response https://doi.org/10.7554/eLife.70494.sa2

## Additional files

### Supplementary files
• Supplementary file 1. Preliminaries and additional results for phylogeography, population genetics, species delimitation and ecological niche modelling analyses.

• Supplementary file 2. The references and notes for all citations in Supplementary file 1A and Supplementary file 1J.

• Transparent reporting form

### Data availability
All data generated or analysed during this study are either included in the manuscript and supporting files, or submitted online depositories. All Sequences generated in the present study were deposited to the Genbank database [https://www.ncbi.nlm.nih.gov/ 927 genbank/] under the accession number MW081664- MW081847 (*CR*), MW467646-MW467777 (*ND2*), MW489915-MW489964 (*POMC*), MW489986-MW490035 (*RAG-1*), MW507752-MW507780 (*Rho*). Input files in the form of BEAST XML generated for all molecular dating analyses and species delimitation modelling are available from the Mendeley Data repository http://dx.doi.org/10.17632/wdtw6kn2t4.1 (Othman et al., 2021).

The following dataset was generated:

| Author(s) | Year | Dataset title | Dataset URL | Database and Identifier |
|---|---|---|---|---|
| Othman SN, Litvinchuk SN, Maslova I, Dahn H, Messenger KR, Andersen D, Jowers MJ, Kojima Y, Yasumiba K, Chuang MF, Chen YH, Bae Y, Hoti J, Jang Y, Borzée A | 2021 | From Gondwana to the Yellow Sea: evolutionary diversifications of true Toads (*Bufo* sp.) in the Eastern Palearctic and species boundaries revisit for Asian lineages | https://data.mendeley.com/datasets/wdtw6kn2t4/1 | Mendeley Data, 10.17632/wdtw6kn2t4.1 |

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

# Appendix 1

This supplementary text includes the supportive information of materials and methods and results used to build our conclusion, such as the extended population genetic analyses, and the details of calibration points used for molecular dating estimation. We include the assessment of model selections for the species tree topology and species delimitation modeling. We also include the detailed information of all data sets used for the reconstruction of the paleogeographic maps for the molecular dating analyses.

## Materials and methods

### Molecular analyses

Genomic DNA was extracted from tissues and swab samples with the DNeasy Blood and Tissue Kit following the instruction of the manufacturer (Qiagen Group, Hilden, Germany). We amplified DNA through PCRs with a total volume of 20 µl per tube, containing 35–50 ng/µl of template DNA. The final concentration of the other PCR reagents was such as: 0.125 µM for each forward and reverse primer, 1× Ex Taq Buffer (Takara; Shiga, Japan), 0.2 mM of dNTPs Mix (Takara; Shiga, Japan), 1.875 mM of Magnesium Chloride (MgCl$_2$), 0.1 unit/µl of Ex Taq (HR001A, Takara; Shiga, Japan), and double-distilled water added to make up the final volume. All fragments targeted were amplified with the following PCR thermal profiles: (1) 5 min at 95°C for pre-denaturation; a repeated 35 cycles of (2) 1 min at 95°C for denaturation; (3) 30 s at specific temperature for annealing; (4) 1 min at 72°C for elongation. The amplification was ended by a final elongation at 72°C for 5 min. PCR reactions were performed in a SimpliAmp Thermal Cycler (Applied Biosystems, USA). Products were visualized on 1.5% agarose gel loaded with three microliters of PCR products, run on an Agaro-Power System (A-7020, Bioneer; Republic of Korea) and visualized with a Nucleic acid Bioimaging Instrument Blue Illuminator (NeoScience; Republic of Korea) using TopGreen Nucleic Acid 6× Loading Dye (GenomicBase, Republic of Korea). Samples were sent for purification and sequencing for both forward and reverse directions by Cosmogenetech (Cosmogenetech Co., Ltd., Seoul, Republic of Korea). These sequence data have been submitted to the GenBank databases (*Supplementary file 1J*).

### Population genetic analyses

We ran the isolation by distance (IBD) analysis on the mtDNA data set to test for the correlation between genetic and geographical distances using a (*Mantel, 1967*). The statistical significance of the correlation between matrices was assessed with a Mantel randomization test (10,000 permutations). We then determined the genetic differentiation among populations by calculating the F-statistics, and we determined the proportion of genetic variability found among populations (F$_{ST}$), among populations within groups (F$_{SC}$) and among groups (F$_{CT}$). We estimated the fixation index following the random allelic permutation procedures in ARLEQUIN v.3.1 (*Excoffier and Lischer, 2010*), where we assessed the statistical significance with 10,000 permutations tests. To estimate the genetic population structure, we then grouped the samples according to the six segregated clades inferred from our phylogenetic tree from the mtDNA data set, and performed an AMOVA as implemented in the Arlequin package.

### Population clustering

We converted the 44 sequences of concatenated *POMC-RAG-1-Rho* (1030 bp) to SNP format using PSGspider v.2.1.1.5 (*Lischer and Excoffier, 2012*). We then assigned individuals to one of the eight following populations according to the previously determined clades: (1) western Mainland, (2) central Mainland, (3) southeastern Mainland, (4) northeastern Mainland, (5) Korean Peninsula, (6) Amur River Basin (along with the southern Primorsky Kray and the Sakhalin Island), (7) Taiwan Island, and (8) the Japanese Archipelago. Then, we used an admixture model to determine the most likely clusters in our SNP data set using sampling location as prior (LocPrior). We computed the statistic under 100,000 MCMC chains with 1000 burn-in for 15 iterations using STRUCTURE v.2.3.4 (*Pritchard et al., 2000*). Because of the uncertainties in the real number of subpopulations, we set the range of cluster (K) from K=1 to K=10 using the formula of K=1 to K=n+2 (n=number of populations) (*Evanno et al., 2005*; *Janes et al., 2017*). In order to select the best possible number of clusters in the population and interpret the raw results obtained, we further characterized the result obtained using STRUCTURE Harvester (*Earl and vonHoldt, 2012*) and the Clump algorithm implemented in the CLUMPAK server (*Kopelman et al., 2015*). We visualized the results using Structure Plot 2.0 (*Ramasamy et al., 2014*).

## Evaluation of introgression pattern

To test the frequencies of discordancy in our multilocus *POMC-RAG-1-Rho* data set (1030 bp), we tested for incomplete Lineage Sorting (ILS) or introgression following population admixture of East Asian *Bufo* clades. To do so, we examined the SNP genealogies following the modified gene flow analysis of D-statistic (*Patterson et al., 2012*). We calculated Patterson's D-statistic under the assumption that the rates of substitution are equal between the unlinked loci. We computed the D-statistics using the 'CalcD' function in the Evobir package implemented in cran R (*R Core Team, 2020*) with 1000 replicates per run. In addition, we applied a simple $\chi 2$ variant of the ABBA-BABA test to compare the frequencies of discordant SNP genealogies and trace the pattern of incomplete lineage sorting, based on the rule of equal ratio (50:50) of ABBA-BABA test (*Martin et al., 2015*). This test assumes a strict pattern of evolutionary history in a tree where the substitution rate is always equal and the loci are unlinked (represented by a D-statistics equivalent to 0, where the equal 50:50 ratio of ABBA-BABA test is achieved). In opposition, a D-statistic value significantly different from 0 suggests the presence of introgression in the evolutionary tree, where the event distorted the equal substitution rate along the evolutionary history timeframe.

## Species tree: fossil calibrations process

For the species tree reconstruction, we compared the topology of our species tree based on the global phylogenetic relationship of Bufonidae proposed by *Portik and Papenfuss, 2015*. We calibrated the maximum and minimum age constrained based on a total fossils evidence. Each calibration point set up was described as:

1. A minimum age of 11.0 Mya for the emergence of the *Rhinella* genus in South America based on the oldest fossil of *R. marina* found in Colombia (*Estes and Wassersug, 1963*). We enforced this calibration point under a lognormal distribution with mean in real space =10.0, log SD=1.1, and offset =11.0. This calibration resulted in a credibility interval of 5%–95% that ranged between 11.9 Mya and 44.3 Mya.
2. A minimum age of 18.0 Mya for the origin of toads belonging to the *Bufotes* genus based on the oldest fossil of *B. viridis* sensu lato found in France (*Bailon and Hossini, 1990*). We enforced this calibration point under a lognormal distribution with mean in real space =8.0, log (SD)=0.6, and offset =16.0. This calibration resulted in a credibility interval of 5%–95% that ranged between 18.5 Mya and 33.9 Mya.
3. A minimum age of 7.0 Mya for the origin of *Epidalea* and *Strauchbufo* based on the oldest fossil of *E. calamita* found in Spain (*Sanchiz, 1998*; *Sanchiz, 1997*) and *S. raddei* found in Russia (*Syromyatnikova, 2015*). We enforced this calibration point under a log-normal distribution with mean in real space =6.0, log (SD)=0.8, and offset =6.0. This calibration resulted in a credibility interval of 5%–95% that ranged between 7.17 Mya and 22.2 Mya.
4. A minimum age of 9.2 Mya for the origin of the Western Palearctic *Bufo* group, based on the oldest fossil of *B. bufo* found in the Czech Republic (*Roček and Rage, 2003*). We enforced this calibration point under a lognormal distribution with mean in real space =6.0, log (SD)=0.8, and offset =8.0. This calibration resulted in a credibility interval of 5%–95% that ranged between 9.17 Mya and 24.2 Mya.
5. A minimum age of 1.0 Mya for the origin of the Eastern Palearctic *Bufo* group, based on the oldest recorded fossil of *B. gargarizans* found in Beijing, China (*Ratnikov, 2001*). We enforced this calibration point under a log-normal distribution with mean in real space =3.0, log (SD)=0.8, and offset =0.5. This calibration resulted in a credibility interval of 5%–95% that ranged between 1.08 Mya and 8.62 Mya.

We assembled the species trees from all analyses with LogCombiner v.2.6 (*Bouckaert et al., 2019*). Thereafter the species tree reconstruction, we also assessed and visualized the assembled species trees with DensiTree v.2.0 (*Bouckaert, 2010*).

## Calibrating the time tree of East Asian *Bufo*

For the divergence dating, we first evaluated the most appropriate molecular clock to be used for the data sets. To do so, we tested a strict molecular clock by calculating the likelihood ratio (LR) following the equation: LR =2 ($-Ln_A$ – ($-Ln_0$)), adapted from *McGee, 2002* using PAUP v.4.0. The significant Chi-test for the mtDNA data set (df: 130; p<0.0001; *Supplementary file 1N*) supports a variation in the rates of substitution in our trees, and indicates the impossibility to reject the

global clock hypothesis. As a result, we used of a relaxed molecular clock model for the divergence dates estimates with the mtDNA data set, in agreement with recent studies on amphibians (*Liedtke et al., 2016*). In opposition, the homogeneity test accepted the hypothesis stating that our nuDNA data evolved according to a strict molecular clock, supported by a non-significant Chi-test (df: 42; p=0.019; *Supplementary file 1N*). Hence, we employed a strict clock model for the nuDNA data set. Here, we calibrated the time tree for the East Asian *Bufo* lineage based on the primary calibration (fossils evidence) and past ecological events, then followed by the secondary calibrations obtained from *CR* mitochondrial estimation in the literatures. The seven internal calibration points are:

1. A minimum age of 6.0 Mya (Late Miocene; *Kamata and Kodama, 1994*) to a maximum age of 10.0 Mya (Middle Miocene; *Taira, 2001*) for the isolation of the Japanese *Bufo: B. j. formosus* and *B. j. japonicus*. The time range is based on the age of the drift of the Japanese plates away from Eurasia. We enforced this calibration point under a normal distribution with mean =1.0, sigma (σ)=1.0, and offset =7.0, resulting in a credibility interval of 5%–95% that ranged between 6.36 Mya and 9.64 Mya.

2. A minimum age of 5.0 Mya to a maximum age of 10.0 Mya for the emergence TMRCA of Asian mainland *Bufo* (*B. tibetanus*) in western China (*Macey et al., 1998*), matching with an active uplifting phase of the QTP during the Miocene (*Zhang et al., 2016*). We enforced this calibration under a normal distribution with mean =1.0, σ=1.5, and offset =7.0, resulting in a credibility interval of 5%–95% that ranged between 6.53 Mya and 10.5 Mya.

3. A minimum age of 2.7 Mya to a maximum age of 6.2 Mya for the estimated independent origin of *B. stejnegeri* in Northeast Asia (Late Miocene – Early Pliocene: c. 4.3 Mya; *Fong et al., 2020*). We enforced this secondary calibration under a normal distribution with mean =1.0, σ=1.0, and offset =3.5, resulting in a credibility interval of 5% to 95% that range between 2.86 Mya and 6.14 Mya.

4. A minimum age of 2.5 Mya to a maximum age of 4.0 Mya for the split between *B. andrewsi* and *B. gargarizans* present on highlands and *B. gargarizans* present on lowland of the southwestern Asian mainland (*Fu et al., 2005*; *Zhan and Fu, 2011*). This calibration point matches with the Plio-Pleistocene uplift of the QTP (*Lei et al., 2014*). We enforced this secondary calibration under a normal distribution with mean =0.3, σ=0.4, and offset = 2.5, resulting in a credibility interval of 5%–95% that ranged between 2.14 Mya and 3.46 Mya.

5. A minimum age of 0.65 Mya to a maximum age of 1.35 Mya for the emergence of *B. bankorensis* on the Taiwanese Island, originating from a single colonization even by *B. gargarizans* from the Asian mainland (*Yu et al., 2014*). We linked this calibration point with the oldest related fossil: *B. gargarizans miyakonis* in Miyako Island from the Ryukyu archipelago (Late Pleistocene: <1.0 Mya) (*Nokariya and Hasegawa, 1985*). We enforced this calibration under a lognormal distribution with mean =1.0, log (SD)=0.3, and offset =0.05, resulting in a credibility interval of 5%–95% that ranged between 0.63 Mya and 1.62 Mya.

6. A minimum age of 0.7 Mya for the only recorded fossil assigned to '*Bufo*' (presumed *B. gargarizans*) found in Beijing (*Ratnikov, 2001*). We enforced this calibration on the *B. gargarizans* monophyletic clade occurring on the southeastern and northeastern Asian mainland under a lognormal distribution with mean =0.5, log (SD)=1.0, and offset =0.5, resulting in a credibility interval of 5%–95% that ranged between 0.56 Mya and 2.07 Mya.

7. A minimum age of 0.8 Mya to a maximum age of 2.0 Mya for the Pleistocene refugium and isolation of *B. gargarizans* on the Korean Peninsula, originating from the southwestern mainland and followed by the colonization to northern latitudes (Pleistocene; 2.0–0.8 Mya) (*Borzée et al., 2017*). We enforced this secondary calibration on the southwestern and septentrional East Asian clades of *B. gargarizans* under a normal distribution with mean =0.4, σ=0.3, and offset =1.0, resulting in a credibility interval of 5%–95% that ranged between 0.91 Mya and 1.89 Mya.

## Results

### Molecular phylogeny of East Asian *Bufo*

Among the six clades recovered by the concatenated *CR-ND2*, we found two distinctive clades within the Japanese *Bufo* corresponding to the subspecies *B. j. formosus* (ML: 97.8/100, PP: 0.98; Clade 1 on *Figure 4a*) and the subspecies *B. j. japonicus* (ML: 99.5/100, PP: 1.0; Clade 2 on *Figure 4B*). Clade

3 represented *B. stejnegeri* on the Korean Peninsula (ML: 80.1/100, PP: 1.0; *Figure 4B*). Clade 4 (ML: 97.6/100, PP: 0.5; *Figure 4B*) consisted of *B. tibetanus* (ML: 99.6/100, PP: 0.98; *Figure 4B*) and its sister clade, *B. andrewsi* (ML: 100/100, PP: 1.0; *Figure 4B*), both distributed in Western Mainland Asia. Clade 5 included the populations of *B. gargarizans* restricted to central of East Asia (ML: 90.8/100, PP: 0.5; *Figure 4B*), containing sister clades of *B. gargarizans* in the central Mainland (ML: 93.1/100, PP: 0.7; *Figure 4B*), *B. bankorensis* in Taiwan Island, *B. gargarizans* in southeastern Mainland Asia (ML: 87.6/100, PP: 0.92; *Figure 4B*), and another *B. gargarizans* in northeastern Mainland Asia (ML: 89.9/100, PP: 0.93; *Figure 4B*). Clade 6 recovered *B. gargarizans* that was geographically restricted to septentrional East Asia (Clade 6: ML: 96.8/100, PP: 0.99; *Figure 4B*): contained a subclade of *Bufo* distributed in Amur River Basin (ML: 93.9/100, PP: 1.0; *Figure 4B*) and a subclade distributed in Korean Peninsula (ML: 59.2/100, PP: 1.0; *Figure 4B*).

Whereas, the six clades recovered from nuDNA tree clustered *B. gargarizans* restricted to the southeastern and northeastern Asian Mainland (ML: 78.9/100; PP: 0.73; Clade 1; *Figure 4D*). The second clade contained of *B. gargarizans* distributed in Central Mainland Asia (ML: 79.1/100; PP: 0.94; Clade 2; *Figure 4D*). The third clade nested the two Korean *Bufo* clades, supporting the placement of *B. stejnegeri* and the Korean *B. gargarizans* in a clade (ML: 75.5/100; PP: 0.77; Clade 3; *Figure 4D*). The same clade also assigned *B. gargarizans* distributed in southeastern Mainland Asia, *B. bankorensis* distributed in Taiwan Island (ML: 93.4/100; PP: 1.0; Clade 3; *Figure 4D*) and a subspecies of *B. gargarizans* distributed in the Amur River Basin (ML: 79.6/100; PP: 0.70; Clade 3; *Figure 4D*). We also recovered a clade consisting of the Japanese *Bufo* clade (ML: 75.5/100; PP: 0.69; Clade 4; *Figure 4D*), and a monophyletic clade of *B. gargarizans* geographically restricted to the northeastern Mainland (ML: 86/100; PP: 0.70; Clade 5; *Figure 4D*). Finally, Clade 6 recovered another *B. gargarizans* of the mainland group restrictly distributed across western, central, southeastern, and northeastern Asia (ML: 84.5/100; PP: 0.75; *Figure 4D*).

## Appendix 1—key resources table

| Reagent type (species) or resource | Designation | Source or reference | Identifiers | Additional information |
|---|---|---|---|---|
| Biological sample (*Bufo gargarizans*) | Buccal DNA swabs and road kills tissues of *Bufo* individuals across East Asia. | GenBank, this study | Voucher list in *Supplementary file 1J* | https://www.ncbi.nlm.nih.gov/genbank/ |
| Biological sample (*B. andrewsi*) | Buccal DNA swabs and road kills tissues of *Bufo* individuals across East Asia. | GenBank, this study | Voucher list in *Supplementary file 1J* | https://www.ncbi.nlm.nih.gov/genbank/ |
| Biological sample (*B. bankorensis*) | Buccal DNA swabs and road kills tissues of *Bufo* individuals across East Asia. | GenBank, this study | Voucher list in *Supplementary file 1J* | https://www.ncbi.nlm.nih.gov/genbank/ |
| Biological sample (*B. stejnegeri*) | Buccal DNA swabs and road kills tissues of *Bufo* individuals across East Asia | GenBank, this study | Voucher list in *Supplementary file 1J* | https://www.ncbi.nlm.nih.gov/genbank/ |
| Biological sample (*B. japonicus formosus*) | Buccal DNA swabs and road kills tissues of *Bufo* individuals across East Asia | GenBank, this study | Voucher list in *Supplementary file 1J* | https://www.ncbi.nlm.nih.gov/genbank/ |
| Biological sample (*B. j. japonicus*) | Buccal DNA swabs and road kills tissues of *Bufo* individuals across East Asia | GenBank, this study | Voucher list in *Supplementary file 1J* | https://www.ncbi.nlm.nih.gov/genbank/ |
| Biological sample (*B. s. sachalinensis*) | Buccal DNA swabs and road kills tissues of *Bufo* individuals across East Asia | GenBank, this study | Voucher list in *Supplementary file 1J* | https://www.ncbi.nlm.nih.gov/genbank/ |
| Gene (*B. gargarizans*) | Multi loci (*CR-ND2-POMC-RAG-1-CRCX4-Rho*) | GenBank, this study | Voucher list in *Supplementary file 1J* | https://www.ncbi.nlm.nih.gov/genbank/ |
| Gene (*B. andrewsi*) | Multi loci (*CR-ND2-POMC-RAG-1-CRCX4-Rho*) | GenBank, this study | Voucher list in *Supplementary file 1J* | https://www.ncbi.nlm.nih.gov/genbank/ |

*Appendix 1 Continued on next page*

*Appendix 1 Continued*

| Reagent type (species) or resource | Designation | Source or reference | Identifiers | Additional information |
|---|---|---|---|---|
| Gene (*B. bankorensis*) | Multi loci (*CR-ND2-POMC-RAG-1-CRCX4-Rho*) | GenBank, this study | Voucher list in **Supplementary file 1J** | https://www.ncbi.nlm.nih.gov/genbank/ |
| Gene (*B. stejnegeri*) | Multi loci (*CR-ND2-POMC-RAG-1-CRCX4-Rho*) | GenBank, this study | Voucher list in **Supplementary file 1J** | https://www.ncbi.nlm.nih.gov/genbank/ |
| Gene (*B. j. formosus*) | Multi loci (*CR-ND2-POMC-RAG-1-CRCX4-Rho*) | GenBank, this study | Voucher list in **Supplementary file 1J** | https://www.ncbi.nlm.nih.gov/genbank/ |
| Gene (*B. j. japonicus*) | Multi loci (*CR-ND2-POMC-RAG-1-CRCX4-Rho*) | GenBank, this study | Voucher list in **Supplementary file 1J** | https://www.ncbi.nlm.nih.gov/genbank/ |
| Gene (*B. s. sachalinensis*) | Multi loci (*CR-ND2-POMC-RAG-1-CRCX4-Rho*) | GenBank, this study | Voucher list in **Supplementary file 1J** | https://www.ncbi.nlm.nih.gov/genbank/ |
| Commercial assay or kit | Qiagen DNeasy Blood and Tissue Kit | QIAGEN Group, Hilden, Germany | Cat. no./ID: 69504 | |
| Commercial assay or kit | PCR purification CaspaseGlo 3/7 | PROMEGA | G8090 | |
| Commercial assay or kit | BigDye Terminator v.3.1 Cycle Sequencing Kit | Cosmo Genetech Corp. (Seoul, Republic of Korea) | Thermo Fisher Scientific, Gangnam, Republic of Korea | |
| Sequence-based reagent | Control B-H | Goebler et al. (1999) | PCR primers for control region (*CR*); **Supplementary file 1K** | Forward: GTCCATTGGAGGTTAAGATCTACCA |
| Sequence-based reagent | CR-BGarF and CR-BGarR | **Borzée et al., 2017** | PCR primers for control region (*CR*); **Supplementary file 1K** | Forward: TTGGACGATAGCAAGGAACACTC Reverse: CCTGACTTCTCTGAGGCCGCTTT |
| Sequence-based reagent | conBG-L and conBG-H | **Liu et al., 2000** | PCR primers for control region (*CR*); **Supplementary file 1K** | Forward: GCACGATAGCAAGGAACAC Reverse: CCGCTTTAAGGTACGATA |
| Sequence-based reagent | ND1-L-int | **Fu et al., 2005** | PCR primers for NADH dehydrogenase 1 (*ND1*) and NADH dehydrogenase 2 (*ND2*); **Supplementary file 1K** | Forward: CGAGCATCC TACCCACGATTTCG |
| Sequence-based reagent | ND1-H4980 | **Macey et al., 1998** | PCR primers for NADH dehydrogenase 1 (*ND1*) and NADH dehydrogenase 2 (*ND2*); **Supplementary file 1K** | Reverse: ATT TTTCGTAGTTGGGTTTGRTT |
| Sequence-based reagent | BGND2F and BGND2R | This study | PCR primers for *ND2*; **Supplementary file 1K** | Forward: TCTCATTCCCAATCTCACTTCTACT Reverse: GCC TCACCCTCCGACAATA |
| Sequence-based reagent | POMC DRV F1 and POMC DRV R1 | **Vieites et al., 2009** | PCR primers for proopiomelanocortin (*POMC*); **Supplementary file 1K** | Forward: ATATGTCATGASCCAYTTYCGCTGGAA Reverse: GGCRTTYTTGAAWAGAGTCATTAGWGG |
| Sequenced-based reagent | snoBGRAG1F and snoBGRAG1R | This study | PCR primers for recombination activating gene 1 (*RAG-1*); **Supplementary file 1K** | Forward: TGAGAAACGCAGAGAAAGCCC Reverse: GACGGGTGGCATCACAAAGAG |
| Sequence-based reagent | BGRho01-F and BGRho01-R | This study | PCR primers for rhodopsin (*Rho*); **Supplementary file 1K** | Forward: CGACTACACCCTGAAGCC Reverse: CCAACAGATAAGGAAGAAGACCAC |
| Chemical compound, drug | Ethyl alcohol anhydrous 94.5%–99.9% | DAEJUNG | (EP/GR) 500 ml/1 L; CAS: 64-17-5 | |
| Software, algorithm | BEAST v.2.6.1 and STARBEAST | **Bouckaert et al., 2019** | https://www.beast2.org/ | |

*Appendix 1 Continued*

| Reagent type (species) or resource | Designation | Source or reference | Identifiers | Additional information |
|---|---|---|---|---|
| Software, algorithm | SNP and AFLP Package for Phylogenetic analysis (SNAPP) in BEAST v.2.6.1 | *Grummer et al., 2014*; *Bouckaert et al., 2019* | https://www.beast2.org/snapp/ | |
| Software, algorithm | QGIS v.10.1. | ESRI, CA | | |
| Software, algorithm | ArcMap v. 10.6 | ESRI, Redlands, CA | | |
| Software, algorithm | MAXENT v.3.1 | http://www.cs.princeton.edu/~schapire/maxent/; Phillips et al. 2004, 2006 | | |
| Software, algorithm | R version 3.5.1 and | R Core Team (2017) | | |
| Software, algorithm | R Studio version 1.1.442 | R Studio Team (2020) | | |
| Software, algorithm | MIGRATE v.3.6.11 | *Beerli et al., 2019* | https://peterbeerli.com/migrate-html5/ | |
| Software, algorithm | MrBayes is 3.2.7 | *Huelsenbeck and Ronquist, 2001* | | |

