## [Editor Report]

Othman et al., resolve the phylogeography of Bufo at global, continental and regional scales. The strengths of the paper reside in the extreme detail, the inclusive nature of the ideas, and insightful reconstructions of the evolutionary history of Bufo species. Furthermore, taxonomic treatments are a strong part and a nice addition of the paper. The manuscript can be of interest to herpetologists, but also biogeographers and evolutionary biologists interested in the geologic history of Asia.

---

## [Decision Letter]

**Decision letter after peer review:**

Thank you for submitting your article "From Gondwana to the Yellow Sea: evolutionary diversifications of true toads (Bufo sp.) in the Eastern Palearctic, and a revisit of species boundaries for Asian lineages" for consideration by *eLife*. Your article has been reviewed by 2 peer reviewers, and the evaluation has been overseen by a Reviewing Editor and George Perry as the Senior Editor. The following individual involved in review of your submission has agreed to reveal their identity: Jonathan J Fong (Reviewer #1).

The reviewers have discussed their reviews with one another, and the Reviewing Editor has drafted this to help you prepare a revised submission. Both reviewers found merit in your study, but also both of them found important shortcomings with the style of your writing, and with the amount of detail in some parts of the text. A successful revision will require that you change this, and the main text on the manuscript becomes more readable for general audiences, without losing the precision that specialist readers of *eLife* are looking for.

Essential revisions:

1) It seems we both had the same reading of the manuscript. It can be a great contribution, but the text is rather loaded with unnecessary details; messages are hidden in large tables; figures are nice, but many times just too busy to appreciate; and, a lot of very specific jargon is retained in most sentences. Thus, the reviewers suggest the authors to reorganize the manuscript in a text that is more hypothesis testing, less descriptive. One idea you can try is adding subsections that contain specific evolutionary/ecological hypotheses your manuscript addresses. Another way is through eliminating many sentences that describe numerical results, (analysis) model assumptions, and other sentences that do not say explicitly what they really want to say. For example, replace lines 332-334 (or the whole paragraph for that end) for the text closer to *the frogs have different niches!*. Finally, you can build longer appendices.

2) Please provide actual taxonomic accounts of the species, and justify properly new taxonomic changes. Lines 410-418. Not sure this paragraph is needed since you really did not provide taxonomic treatment of the species. Please include these changes in all sections of the manuscript.

3) On the reviewers added an annotated pdf with detailed suggestions. It is a combination of minor corrections to text, to issues about clarity and structure. Whenever possible, the reviewer has tried to provide some alternative approaches-they should be tried, but ultimately the reviewer trust the authors will decide what works best.*Reviewer #1:*

Summary: This paper looks at the evolution of toad species in the family Bufonidae at two different geographic scales: global and regional in Asia. On the global scale, the authors verify many of the previously known patterns in Bufonidae, such as Asian Bufo being a result of a single event. The regional scale is where much of the new work comes in. On the regional scale, the authors take an exhaustive approach to clarify the systematics of Bufo, using phylogenetic analyses, divergence dating, structure analysis, evaluation of migration and introgression, ecological niche modelling. Through this integrative approach, the authors are able to provide a good assessment of the Bufo species present in Asia, as well as hypotheses for the speciation, and taxonomic revision.

Strengths: The goals are clear, analyses appropriate and thorough, and the results support their discussion. The authors have worked hard to use applicable analyses and test assumptions/priors by modifying them. The manuscript is generally well written-I especially liked the introduction and discussion, with most of it being very clear and straightforward.

Weaknesses: Abstract could be revised to better reflect what the manuscript is about-currently the abstract undersells what they are doing. The analyses are thorough, but the authors could do a better job simplifying and interpreting the results. I think many of the tests of the assumptions/priors could be quickly discussed (e.g. all of our tests point towards the same result) and not distract from their results. A clearer, more straightforward explanation of the results would help clarify the interesting points. The figures are pretty and must have taken a lot of work, but are currently too complicated. A figure should be easily and quickly understood, even without reading the manuscript-currently most of the figures would need at least 30 minutes to fully understand (if possible). As with the results, simplification would benefit the manuscript. I have provided some suggestions, but ultimately it will need to be tested. Some general suggestions for the writing-there are lots of terms (names, regions) and it would help if the terms were reduced and consistent across sections of the paper. Also, with names, the author should be aware that the formatting of the manuscript is a bit different (with the methods at the end). As a result, some of the names suddenly appear for the first time in the results (e.g. many of the B. g subspecies, B. sachalinensis [which is not a commonly used or known name]). I do not know the proper format, but I was actually confused about this because you have two B. s. species (B. sachalinensis and B. stejnegeri). For much of the paper, I didn't know which the abbreviation was referring to. The authors need to be careful to follow proper naming protocol as well as stay consistent throughout the paper.

Appraisal: Yes, the authors achieved their aims and the results support their conclusions.

Impact: For future phylogenetic studies, this paper provides a good foundation on a suite of potential analyses that can be performed, and their uses. For biogeographic studies in Eurasia and East Asia (especially Northeast Asia), this study provides a nice study which to compare to. For people interested in Asian Bufo species, this paper provides a good synthesis and revision of the species in the region There have been occasional studies comparing the species that often have conflicting results (e.g. a single or multiple species). This study combines multiple data types (genetics, geography, niche, geology) to try to clarify the situation. It is a reasonable conclusion-don't know if it is the "true" answer, but it is a great synthesis of what we currently know, and future researchers can use as a comparison point.*Reviewer #2:*

Othman et al., merge a large number of analyses to test for phytogeographic reconstruction of Bufo toads across East Asia. To do so they resolve both (1) the history of the genus since Gondwana split (reconstructing the evolution of the clade in the Americas, North Africa, and Eurasia) and (2) supply Eastern Palearctic reconstructions of 'local' Bufo species to test among competing hypotheses of regional Miocene radiations.

The strengths of the paper reside in the extreme detail, the inclusive nature of the ideas, and insightful reconstructions of the evolutionary history of these Bufo species, as supported by a large and complementary set of analyses. The weaknesses are actually the same! as the text is rather difficult to follow (specially for non-specialists), the main messages are hidden among many pages of (numerical) results, and large amounts of information are simplified in impressive, but many times difficult, figures. The large number of analyses do not really help to retain the take-home message as readers would have to be knowledgeable in many different methodologies to fully understand the whole picture. One further weakness is the taxonomic treatment of the species. In my opinion, taxonomic changes should be done in a more traditional way, justifying, species by species, the changes proposed in a proper and formal way. In a manuscript like this one, authors should marry the changes they propose since the beginning! This way older names are not perpetuated by the same manuscript that is trying to resolve them. Resolving taxonomy means that manuscript/figures/tables should be devoid of such names as *B. sachalinensis spp B. sachalinensis sachalinensis*, complexes, and other terminology.

---

## [Author Response]

Essential revisions:1) It seems we both had the same reading of the manuscript. It can be a great contribution, but the text is rather loaded with unnecessary details; messages are hidden in large tables; figures are nice, but many times just too busy to appreciate; and, a lot of very specific jargon is retained in most sentences. Thus, the reviewers suggest the authors to reorganize the manuscript in a text that is more hypothesis testing, less descriptive. One idea you can try is adding subsections that contain specific evolutionary/ecological hypotheses your manuscript addresses. Another way is through eliminating many sentences that describe numerical results, (analysis) model assumptions, and other sentences that do not say explicitly what they really want to say. For example, replace lines 332-334 (or the whole paragraph for that end) for the text closer to the frogs have different niches! Finally, you can build longer appendices.

We agree that a hypothesis-driven study will be more appealing and engaging for the readers, and we have revised the manuscript accordingly. We have rearranged the content of this study into two sections. We concentrate section 1 on Biogeography of Holarctic bufonids with the goal to refine the time estimates of the split between clades of Holarctic bufonids, in coherence with fossil data. In section 2, we focused on reconstructing the historical biogeography of the Eastern Palearctic *Bufo* and resolving the unclear taxonomic boundaries in the species complex. The new framework of this study is now explained in the modified version of Figure 1 where we explained the key methods involved in both sections to reach our conclusions.

Regarding the technical aspect, we minimized the use of jargons and abbreviations in the main text to communicate better with readers. We maintained the use of jargons, geographical region and scientific terminology where unavoidable, but in this case, we defined the words for readers to understand them. The references are now formatted to match with *eLife*. We included a key resource table at the beginning of material and methods section. The table now contained the information related to the laboratory bench work and bioinformatics programs involved in this study.

Content-wise, here we provide a short summary of the changes we made in each section to address the comments of the editor. Please see the answer to authors for the details.

Abstract: we rephrased the manuscript to follow the changes, and especially the fact that this study is now hypothesis-driven and it has been divided into two sections: Biogeography of Holarctic bufonids and historical biogeography and taxonomy of the Eastern Palearctic *Bufo*

Introduction: We modified the section about biogeography to stress the necessity for comparative hypotheses testing, and introduce the background for our hypotheses.

Results: Our findings are not represented in the two sections mentioned above. The heading of each subtopic is now matching to the headings in the Materials and methods. For each subtopic, we provide a general summary with a simpler wording to explain the pattern showed by the results before presenting the numerical and descriptive data. The numerical results complicating the understanding, such as the descriptive results about the clades recovered by the phylogeny (Lines 305-344), have been shifted to the Appendix 1 (Lines 1868-1896). The numerical results in genetic diversity (Lines 267-268), population clusters (Lines 352-361), population expansion (Lines 434-437) and population migration (Lines 471-490) have been reduced in the main text and the details are referred to their respective Supplementary Tables 4, 6 and 8 in supplementary file 2. Only when unavoidable, such as the node ages for molecular dating estimates (Lines 217-234; 393-429) have to be retained as numerical results to help explain the pattern of speciation, but also because they are essential to the targeted readers of the manuscript. We however removed the 95% highest posterior density (HPD) for each node age and referred the values to Tables 3 in the main text and Supplementary file 1- Supplementary file 1G.

Material and methods: We ensured the consistency in the title and content for section 1 and section 2 throughout the main text. The headings of each analysis in the subsections (Lines 842, 868, 869, 900, 954, 978, 1017, 1035) now matches with subsections in the results (Lines 206, 210, 238, 243, 265, 345, 362, 376, 430, 445, 461, 491).

2) Please provide actual taxonomic accounts of the species, and justify properly new taxonomic changes. Lines 410-418. Not sure this paragraph is needed since you really did not provide taxonomic treatment of the species. Please include these changes in all sections of the manuscript.

According to the International Commission on Zoological Nomenclature (ICZN), the statement of Life Science Identifiers (LSID) availability (in the lines 410-418 of the original draft) is necessary to support the terminology we used for the taxonomic unit *B. sachalinensis*, despite the fact that we did not describe any new species. We therefore retained the statement but shortened the sentence as much as possible to: “*Bufo sachalinensis* (Life Science identifier (LSID*)*: urn:lsid:zoobank.org:pub:14938673-9CEB-4325-BDBC-83A0AC85B8A7).” (Line 540-541), and we shifted the rest of the statement to the data availability statement (Lines 1103-1111). Unfortunately, we were unable to include the actual taxonomic account at this stage because the LSID code for the species can be registered in the ICZN database only after acceptance of manuscript.

3) On the reviewers added an annotated pdf with detailed suggestions. It is a combination of minor corrections to text, to issues about clarity and structure. Whenever possible, the reviewer has tried to provide some alternative approaches-they should be tried, but ultimately the reviewer trust the authors will decide what works best.

We have revised the main text, figures, tables and supporting materials based on the in-depth recommendations of reviewer.

Reviewer #1:Summary: This paper looks at the evolution of toad species in the family Bufonidae at two different geographic scales: global and regional in Asia. On the global scale, the authors verify many of the previously known patterns in Bufonidae, such as Asian Bufo being a result of a single event. The regional scale is where much of the new work comes in. On the regional scale, the authors take an exhaustive approach to clarify the systematics of Bufo, using phylogenetic analyses, divergence dating, structure analysis, evaluation of migration and introgression, ecological niche modelling. Through this integrative approach, the authors are able to provide a good assessment of the Bufo species present in Asia, as well as hypotheses for the speciation, and taxonomic revision.Strengths: The goals are clear, analyses appropriate and thorough, and the results support their discussion. The authors have worked hard to use applicable analyses and test assumptions/priors by modifying them. The manuscript is generally well written-I especially liked the introduction and discussion, with most of it being very clear and straightforward.Weaknesses: Abstract could be revised to better reflect what the manuscript is about-currently the abstract undersells what they are doing. The analyses are thorough, but the authors could do a better job simplifying and interpreting the results. I think many of the tests of the assumptions/priors could be quickly discussed (e.g. all of our tests point towards the same result) and not distract from their results. A clearer, more straightforward explanation of the results would help clarify the interesting points. The figures are pretty and must have taken a lot of work, but are currently too complicated. A figure should be easily and quickly understood, even without reading the manuscript-currently most of the figures would need at least 30 minutes to fully understand (if possible). As with the results, simplification would benefit the manuscript. I have provided some suggestions, but ultimately it will need to be tested. Some general suggestions for the writing-there are lots of terms (names, regions) and it would help if the terms were reduced and consistent across sections of the paper. Also, with names, the author should be aware that the formatting of the manuscript is a bit different (with the methods at the end). As a result, some of the names suddenly appear for the first time in the results (e.g. many of the B. g subspecies, B. sachalinensis [which is not a commonly used or known name]). I do not know the proper format, but I was actually confused about this because you have two B. s. species (B. sachalinensis and B. stejnegeri). For much of the paper, I didn't know which the abbreviation was referring to. The authors need to be careful to follow proper naming protocol as well as stay consistent throughout the paper.Appraisal: Yes, the authors achieved their aims and the results support their conclusions.

Thank you very much for the reviewer’s critical review. We appreciate the reviewer time and efforts to provide constructive criticism on our manuscript. We tackled all the concerns and comments raised as best as possible.

Regarding the comments about the weaknesses, we improved each section and provided explanations point by point below. For the abstract, the very limited length (150 words) has hindered our efforts to narrate our story in details. However, we do agree with reviewer’s recommendation and we clarified the problem statement at the beginning of the abstract to match with the goals of our study.

“Taxa with vast distribution ranges often display unresolved phylogeographic structures and unclear taxonomic boundaries resulting into hidden diversity.” (Lines 45-46).

We also stated the main approach of our study in the abstract through hypotheses, clarifying the reconstruction of Holarctic bufonids biogeography and the reconstruction of evolutionary diversifications in East Asian *Bufo* clades.

“This hypothesis-driven study reveals the evolutionary history of Bufonidae, from the phylogeographic patterns found in Holarctic bufonids from the West Gondwana up to the phylogenetic taxonomy of Asiatic true toads in the Eastern Palearctic” (Lines 46-50).

Finally, we added some details about our taxonomic findings. We stated the implication of our findings, in which we resolved the unclear taxonomic boundaries in *B. gargarizans* complex, and delimited the taxonomic units. In the main text the changes made in the abstract are:

“Our findings reveal a climate-driven adaptation in septentrional Eastern Asian *Bufo*, explained its range shifts towards northern latitudes. We resolve species boundaries within the Eastern Palearctic *Bufo*, and redefine the taxonomic and conservation units of the northeastern species: *B. sachalinensis* and its subspecies”. (Lines 57-62).

To match with this format and include the readers’ perspectives, we have defined the terminologies, geographical regions, scientific names and specific analyses on the first use of the words. A brief introduction to the specific terminologies such as “*Bufo sachalinensis*”, species “complex” and “septentrional” Eastern Asia can be found in the main text as:

“Comparable to the Western *Bufo*, two or more species of Eastern Palearctic *Bufo* form species complexes with similar morphologies similarity and unclear taxonomic boundaries, such as the *B. gargarizans* and *B. japonicus* complexes (Matsui, 1986; Zhan and Fu, 2011; Arntzen et al., 2013).” (Lines 107-110).

To connect our findings, we divided the main text into two sections, both driven by hypotheses. We standardised each subtopic in the results, matched with that of the material and methods as recommended.

In the main text the section added have been written as:

“Our study addresses biogeographic scenarios explained in two different sections: (1) the biogeography of Holarctic bufonids and, (2) the biogeography of Eastern Palearctic *Bufo* and the taxonomic revision of the species complexes.

Section 1: Biogeography of Holarctic bufonids

The goal was to refine the time estimates of the split between clades of Holarctic bufonids, in coherence with fossil data (Figure 2, Figure 2—figure supplement 1 and Supplementary Table 2).” (Lines 203-209)

“Section 2: Evolutionary diversification of Eastern Palearctic *Bufo*

Here, we focused on reconstructing the historical biogeography of Eastern Palearctic *Bufo* and resolving the taxonomic boundaries of the species complex within the genus. The vast distribution and taxonomic inconsistencies in Eastern Palearctic *Bufo* warrant a careful examination of the hypotheses proposed by previous studies.” (Lines 238-242)

Regarding the point on binomial names, we have standardised the use of the binomial names for each species, including subspecies, such as *B. sachalinensis sachalinensis* (Amur River Basin clade of “*B. gargarizans*”) and its subunit, *Bufo sachalinensis* cf. *sachalinensis* (previously known as “*B. gargarizans*” from the Korean Peninsula) following the International Code of Zoological Nomenclature. We have defined the species and its range in the main text to avoid such confusion especially between Korean *Bufo*, *B. stejnegeri* and *B. sachalinensis* (Lines 284-287). Finally, we revised the terminology use between *Bufo* members in East Asia, and matched the names in the main text with all our figures and tables.

Impact: For future phylogenetic studies, this paper provides a good foundation on a suite of potential analyses that can be performed, and their uses. For biogeographic studies in Eurasia and East Asia (especially Northeast Asia), this study provides a nice study which to compare to. For people interested in Asian Bufo species, this paper provides a good synthesis and revision of the species in the region There have been occasional studies comparing the species that often have conflicting results (e.g. a single or multiple species). This study combines multiple data types (genetics, geography, niche, geology) to try to clarify the situation. It is a reasonable conclusion-don't know if it is the "true" answer, but it is a great synthesis of what we currently know, and future researchers can use as a comparison point.Reviewer #2:Othman et al., merge a large number of analyses to test for phytogeographic reconstruction of Bufo toads across East Asia. To do so they resolve both (1) the history of the genus since Gondwana split (reconstructing the evolution of the clade in the Americas, North Africa, and Eurasia) and (2) supply Eastern Palearctic reconstructions of 'local' Bufo species to test among competing hypotheses of regional Miocene radiations.The strengths of the paper reside in the extreme detail, the inclusive nature of the ideas, and insightful reconstructions of the evolutionary history of these Bufo species, as supported by a large and complementary set of analyses. The weaknesses are actually the same! as the text is rather difficult to follow (specially for non-specialists), the main messages are hidden among many pages of (numerical) results, and large amounts of information are simplified in impressive, but many times difficult, figures. The large number of analyses do not really help to retain the take-home message as readers would have to be knowledgeable in many different methodologies to fully understand the whole picture. One further weakness is the taxonomic treatment of the species. In my opinion, taxonomic changes should be done in a more traditional way, justifying, species by species, the changes proposed in a proper and formal way. In a manuscript like this one, authors should marry the changes they propose since the beginning! This way older names are not perpetuated by the same manuscript that is trying to resolve them. Resolving taxonomy means that manuscript/figures/tables should be devoid of such names as B. sachalinensis spp B. sachalinensis sachalinensis, complexes, and other terminology.

Thank you very much for the reviewer’s time in revising our manuscript. Based on the general comments, we noted the concerns of the reviewer regarding the complexity of our study, to which we agree as it involves diverse analyses. We modified the text to simplify the message and help reader’s comprehensions.

We restructured the manuscript to fit a hypothesis testing approach, and based on the editor and reviewer’s recommendations, we divided the manuscript into two main sections, section 1 focusing on the biogeography of Holarctic bufonids and section 2 concentrating on biogeography and taxonomy of Eastern Palearctic *Bufo*.

To clarify the presentation of the results, we moved some of the descriptive statistic and numbers to the supplementary materials, added a general explanation on the pattern of results and their implication to our conclusions. We also improved the figure presentations, added panel in the figures, and simplified the description of each figure to help the readers’ understanding.

Regarding the taxonomic treatment and terminology, we followed the Zoological nomenclature code for the names used in our taxonomic redefinition on *B. sachalinensis* species. As per the rules of the code, we will register a taxonomic act for the use of this terminology with Zoobank once the paper is accepted.

As an explanation, the epithet “*B. sachalinensis sachalinensis*” represents a *Bufo* species distributed in the Amur River Basin, and *B. sachalinensis* stems from the original description of the species, supported by its presence at the type locality and the description of its holotypes (the details on the species description are provided in Supplementary Table 1 and Figure 1—figure supplement 1).

We corrected the name *Bufo sachalinensis* spp. and use epithet “*Bufo sachalinensis* cf. *sachalinensis”* to represent a discrete monophyletic subclade within *B. sachalinensis* restricted to Korean Peninsula. The distinction of this Korean subclade from *B. sachalinensis sachalinensis* was supported by species delimitation model, and it’s crucial to conserve the genetic diversity of this Korean subclade. Hence, we proposed a conservation unit for this subclade, and naming it permissible following the taxonomic code such as “*B. sachalinensis cf. sachalinensis*”.

To avoid further confusion for the readers, we standardised the use *B. sachalinensis* as the terminology of reference in the manuscript, and we matched the terminology in the figures and tables.

References:

Barnes GL. 2003. Origins of the Japanese Islands: The New “Big Picture.” *Japan Review* 15:3–50. doi:10.2307/25791268

Bell RC, Irian CG. 2019. Phenotypic and genetic divergence in reed frogs across a mosaic hybrid zone on São Tomé Island. *Biological Journal of the Linnean Society* 128:672–680. doi:10.1093/biolinnean/blz131

Borzée A, Santos JL, Sánchez-RamÍrez S, Bae Y, Heo K, Jang Y, Jowers MJ. 2017. Phylogeographic and population insights of the Asian common toad (*Bufo gargarizans*) in Korea and China: population isolation and expansions as response to the ice ages. *PeerJ* 5:e4044. doi:10.7717/peerj.4044

Fu J, Weadick CJ, Zeng X, Wang Y, Liu Z, Zheng Y, Li C, Hu Y. 2005. Phylogeographic analysis of the *Bufo gargarizans* species complex: A revisit. *Molecular Phylogenetics and Evolution* 37:202–213. doi:10.1016/j.ympev.2005.03.023

Guo ZT, Ruddiman WF, Hao QZ, Wu HB, Qiao YS, Zhu RX, Peng SZ, J.J.Wei, Yuan BY, Liu TS. 2002. Onset of Asian desertification by 22Myr ago inferred from loess deposits in China. *Nature* 416:159–163.

Leaché A. 2018. Species Trees Estimation with SNAPP : A Tutorial and Example Version , Author information , and Acknowledgements Background Information Programs Used in This Tutorial 1–19.

Lee C, Fong J, Jiang J, Li P, Waldman B, Chong JR, Lee H, Min M. 2020. Phylogeographic study of the *Bufo gargarizans* species complex, with emphasis on Northeast Asia. *Authorea Preprints* 1–14. doi:10.22541/au.160682234.43883694/v1

Macey JR, Schulte JA, Larson A, Fang Z, Wang Y, Tuniyev BS, Papenfuss TJ. 1998. Phylogenetic relationships of toads in the *Bufo bufo* species group from the eastern escarpment of the Tibetan Plateau: a case of vicariance and dispersal. *Molecular Phylogenetics and Evolution* 9:80–87. doi:10.1006/mpev.1997.0440

Xing Y, Ree RH. 2017. Uplift-driven diversification in the Hengduan Mountains, a temperate biodiversity hotspot. Proceedings of the National Academy of Sciences of the United States of America 114:E3444–E3451. doi:10.1073/pnas.1616063114

Yu TL, Lin H Du, Weng CF. 2014. A new phylogeographic pattern of endemic *Bufo bankorensis* in Taiwan Island is attributed to the genetic variation of populations. *PLoS ONE* 9:e98029. doi:10.1371/journal.pone.0098029

Zhan A, Fu J. 2011. Past and present: Phylogeography of the *Bufo gargarizans* species complex inferred from multi-loci allele sequence and frequency data. *Molecular Phylogenetics and Evolution* 61:136–148. doi:10.1016/j.ympev.2011.06.009

Zhan X, Wu Huan, Wu Hong, Wang R, Luo C, Gao B, Chen Z, Li Q. 2020. Metabolites from *Bufo gargarizans* (Cantor, 1842): A review of traditional uses, pharmacological activity, toxicity and quality control. *Journal of Ethnopharmacology* 246:112178. doi:10.1016/j.jep.2019.112178